# Substrate specificity of the TRAMP nuclear surveillance complexes

Clémentine Delan-Forino[1], Christos Spanos [1], Juri Rappsilber [1,2] & David Tollervey [1✉]

During nuclear surveillance in yeast, the RNA exosome functions together with the TRAMP complexes. These include the DEAH-box RNA helicase Mtr4 together with an RNA-binding protein (Air1 or Air2) and a poly(A) polymerase (Trf4 or Trf5). To better determine how RNA substrates are targeted, we analyzed protein and RNA interactions for TRAMP components. Mass spectrometry identified three distinct TRAMP complexes formed in vivo. These complexes preferentially assemble on different classes of transcripts. Unexpectedly, on many substrates, including pre-rRNAs and pre-mRNAs, binding specificity is apparently conferred by Trf4 and Trf5. Clustering of mRNAs by TRAMP association shows co-enrichment for mRNAs with functionally related products, supporting the significance of surveillance in regulating gene expression. We compared binding sites of TRAMP components with multiple nuclear RNA binding proteins, revealing preferential colocalization of subsets of factors. *TRF5* deletion reduces Mtr4 recruitment and increases RNA abundance for mRNAs specifically showing high Trf5 binding.

[1] Wellcome Center for Cell Biology, University of Edinburgh, Kings Buildings, Swann Building, Edinburgh EH9 3BF, UK. [2] Bioanalytics, Institute of Biotechnology, Technische Universität Berlin, 13355 Berlin, Germany. ✉email: D.Tollervey@ed.ac.uk

The transcription, processing, and packaging of eukaryotic RNAs offers many opportunities for errors (reviewed in refs. [1,2]). In consequence, quality control or "surveillance" of aberrant nuclear RNAs is an essential feature of eukaryotic gene expression. In addition, all stable RNA species undergo post-transcriptional maturation, including 3′ processing. A key component of both RNA surveillance and processing pathways is the exosome; an essential, multi-subunit complex that is highly conserved among eukaryotes.

The purified exosome shows weak activity in vitro, indicating that the rapid and processive activity inferred in vivo requires cofactors. In budding yeast, the Trf4/5–Air1/2–Mtr4 poly-adenylation (TRAMP) complexes are major cofactors for the nuclear exosome and critical for nuclear RNA surveillance activity[3–6]. There are variants of the TRAMP complexes, but each has three components: Trf4 or Trf5, Air1 or Air2, and Mtr4, all of which are conserved to humans. Mtr4 is an ATP-dependent RNA helicase[7,8] and is essential for all known activities of the nuclear exosome. In contrast, the other TRAMP components are required for RNA surveillance in the nucleolus and nucleoplasm, but are not known to participate in the accurate processing of stable RNA species. Air1 and Air2 are zinc (Zn)-knuckle, putative RNA-binding proteins[9], whereas Trf4 and Trf5 are noncanonical poly(A) polymerases. Together, they add a short oligo(A) tail, significantly shorter than poly(A) tail added by the canonical polymerase (Pap1)[10]. This is presumed to provide a single-stranded "landing pad" that makes the RNA a better substrate for 3′-end degradation. It was previously suggested that TRAMP recruitment to substrates would be driven by Air1/2 through their RNA-binding activities[4,11] (reviewed in ref. [12]). In addition, Mtr4 includes a distinctive "arch" or "kow" domain, consisting of a β-barrel stalk inserted into the typical DExH core[13–15]. The arch domain is specifically bound by ribosomal biogenesis factors carrying an arch-interacting motif (AIM), enabling direct Mtr4 recruitment to pre-ribosomal RNAs (rRNAs)[16]. The arch was also proposed to function as a docking platform for Trf4 and Air2, acting independently of the helicase activity of the DExH core.

Human cells express at least three complexes that each contain MTR4 and a zinc-finger protein. These include TRAMP, which is predominately nucleolar in humans, as well as NEXT (Nuclear EXosome Targeting) and PAXT (Poly(A) eXosome Targeting) complexes, which are nucleoplasmic[17–19]. Budding yeast appears to lack NEXT and PAXT homologs, but the TRAMP complexes are present in both the nucleolus and nucleoplasm. The Air1/2 and Trf4/5 pairs show some functional redundancy, since the single mutants are viable, whereas double mutants are inviable or severely growth impaired, depending on strain background[4,9,20]. Structural and functional analyses have focused on the Trf4–Air2 TRAMP complex[21–25]. However, both the Air1/2 and Trf4/5 pairs have very significantly diverged in sequence (Fig. 1a, b). The N-terminal regions of Air1 and Air2 are only ~44% identical (amino acids (aa) 1–72 for Air1, from 360 total; aa 1–59 for Air2 from 344 total), while the C termini are substantially different (15% identity between Air1 aa 204–360 and Air2 aa 193–344)[19]. Similarly, Trf4 and Trf5 show homology over their central regions (67% identical aa over 117–519 in Trf4, from 584 total), but diverge in both the N- and C-terminal regions. These sequence divergences are much greater than most duplicated gene pairs in yeast, suggesting that the functions of the different forms of the yeast TRAMP complex may also have diverged and developed distinct specificities in vivo.

The two C-terminal zinc knuckles of Air2 (ZK4 and ZK5, residues 119–199) mediate the interaction with the central domain of Trf4[23,24,26], which requires the presence Air1 or Air2 to adenylate its substrate in vivo[6]. However, a catalytically inactive Trf4 mutant (Trf4-DADA) can support degradation of most Trf4 targets and rescue the lethality of a Δtrf4 Δtrf5 double mutant, indicating that Trf4 can target RNAs to the exosome independently of adenylation[20]. In strains depleted of Mtr4, TRAMP substrates are both stabilized and hyper-adenylated[27]. This shows that Trf4/5 can be recruited to target RNAs and activated independent of Mtr4 and the exosome.

Here, we aimed to determine which TRAMP complexes form in vivo, how they bind to different substrate classes, and how they cooperate with the exosome. To this end, we characterized TRAMP protein–protein and protein–RNA interactions via mass spectrometry (MS) and CRAC (ultraviolet (UV)-cross-linking analyses), respectively. Unexpectedly, the results indicate major roles for Trf4 and Trf5 in TRAMP targeting and recruitment, and a specific role for Trf5 in messenger RNA (mRNA) stability. In contrast, Air1 and Air2 appear to be highly redundant.

## Results

**Three distinct TRAMP complexes are detected in vivo.** The TRAMP complexes potentially comprise four different combinations of Trf4/5 and Air1/2, together with Mtr4. We assessed the actual combinations formed in vivo, by tandem-affinity purification and MS. Air1, Air2, Trf4, and Trf5 were each tagged with His6-TEV-protein A (HTP). Purifications were initially performed under high salt conditions (1 M NaCl) to recover only stable interactions. Associated proteins were identified by MS and subjected to label-free quantification. iBaq scores[28,29] were calculated for each protein recovered (Supplementary Data 2). These indicated that Air1 interacts with both Trf4 and Trf5, while Air2 interacts almost exclusively with Trf4 (Fig. 1c). Previous analyses reported that Mtr4 is not efficiently retained in TRAMP above 125 mM NaCl[4]. Consistent with this, Mtr4 was weakly recovered in the 1 M NaCl preparations. These data demonstrate that three distinct TRAMP complexes co-exist in vivo, containing Trf4 + Air1 (TRAMP4-1), Trf4 + Air2 (TRAMP4-2), and Trf5 + Air1 (TRAMP5-1) (Fig. 1d). Each presumably also associated with Mtr4.

Trf5 and Air2 were not detected in association with either reciprocal precipitation, but it was unclear whether this reflected an inability to interact, or simply a higher affinity for Trf5–Air1 binding. To test this, AIR1 was deleted in the strain expressing Trf5-HTP. In the air1Δ strain, Air2 was well recovered with Trf5-HTP, indicating competition for Trf5 association and redundancy between Air1 and Air2 (Fig. 1c).

To further characterize factors binding to Trf4 and Trf5, these purifications were repeated at 150 mM NaCl, with and without the inclusion of RNase treatment (Supplementary Data 3 and Fig. 1e; colored boxes indicate fold enrichment in the precipitation indicated relative to the non-tagged control; <2-fold enrichment is gray). In this lower stringency purification, Mtr4 was well recovered with both Trf4 and Trf5, independent of RNA. However, in the air1Δ strain, recovery of Mtr4 with Trf5 was RNase sensitive (Trf5 air1Δ). This suggests that the Trf5–Air1 complex binds jointly to Mtr4, and that this interaction is not fully recapitulated by Trf5–Air2, making the complex less stable.

Nab3 and Nrd1, subunits of the Nrd1–Nab3–Sen1 complex, were recovered with Trf4 but not Trf5, consistent with the reported presence of a Nrd1-interacting motif in Trf4 but not Trf5[25]. Both Nab3 and Nrd1 are implicated in RNA surveillance on RNAPII and RNAPIII transcripts, but are not known to participate in pre-rRNA degradation. The 5′ exonuclease Xrn1 showed RNase-sensitive recovery with Trf4 and was also identified with Air1 and Air2. Xrn1 is predominately cytoplasmic, but several studies have reported nuclear roles[30–33], consistent with this finding. Several early binding, pre-mRNA packaging

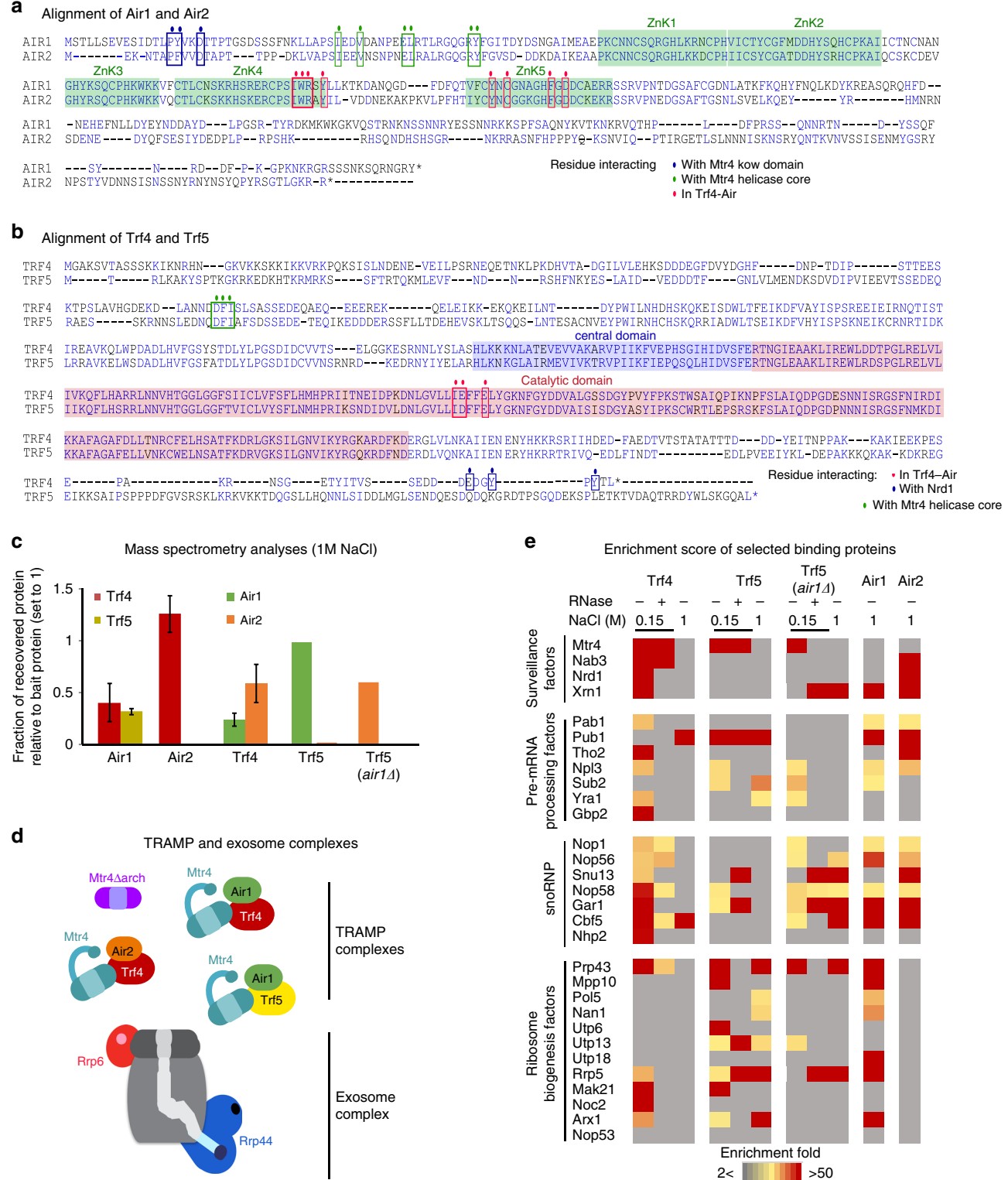

factors were also recovered with Trf4 and Trf5, likely reflecting the role of TRAMP in promoting rapid degradation of non-coding RNAs (ncRNAs)[6,34], and these interactions were largely RNase sensitive.

Multiple proteins associated with both the box C/D and box H/ACA classes of small nucleolar RNAs (snoRNAs) were strongly recovered with Trf4, Air1, and Air2, consistent with the reported involvement of Trf4 in snoRNA maturation[35,36]. Curiously,

recovery of the small nucleolar ribonucleoprotein (snoRNP) proteins with Trf5 was enhanced in the *air1Δ* strain, which has increased Trf5–Air2 association, possibly indicating that Air2 contributes to snoRNP binding. A subset of ribosome synthesis factors was recovered, showing greater interaction with Trf5 than Trf4 and, particularly, with Air1 relative to Air2, for which none showed >2-fold enrichment (Fig. 1e). This suggests a preferential role for TRAMP5-1 in pre-rRNA degradation.

**Fig. 1 Protein interactions involving TRAMP components. a, b** Alignment of paralogs Air1 and Air2 (**a**), Trf4 and Trf5 (**b**). Domains are highlighted with transparent squares (zinc-knuckle domain in green, Trf4 and Trf5 central domain in blue, poly(A) polymerase catalytic domain in red). Residues involved in TRAMP internal interactions are designated by colored circles and squares. **c** Mass spectrometry (LC-MS/MS) analysis of Air1, Air2, Trf4, and Trf5 pull-downs under stringent conditions (1 M NaCl). iBAQ (intensity-based absolute quantification) values were calculated and TRAMP component recovery was represented relatively to bait protein iBAQ set to 1. Two individual replicates were averaged with standard deviation shown as error bars (except for Trf5 where one replicate was excluded due to high signal-to-noise ratio). See Supplementary Data 2 for values. **d** Schematic representation of the TRAMP nuclear cofactor and exosome complexes. Proteins subjected to CRAC in this study are highlighted in color: Mtr4 helicase in dark blue, Mtr4Δarch mutant in purple. Air1 and Air2 zinc-knuckle domains in green and orange, respectively. Trf4 and Trf5 poly(A) polymerase domains in dark red and yellow, respectively. Rrp44 exo- and endonuclease and Rrp6 exonuclease active sites are represented in dark blue and red, respectively. The exosome barrel is colored in gray. **e** riBAQ (relative molar abundances for each protein) was determined by dividing the iBAQ value for each protein by the sum of all non-contaminant iBAQ values. Enrichment was calculated as the riBAQ ratio between TRAMP protein pull-down and the mock sample. All proteins showing a ratio <2 were consid red to be not significantly enriched. See Supplementary Data 3 for values.

Notable omissions were Utp18 and Nop53, which were reported to bind Mtr4 directly in the context of pre-ribosomes and promote TRAMP-independent, pre-RNA processing[16]. The exosome was also very poorly recovered (Supplementary Data 3), consistent with the failure of the original exosome purifications to recover TRAMP components and vice versa[4,5,37]. TRAMP–exosome interactions may be too transient in vivo to be readily recovered.

**The TRAMP complexes exhibit distinct substrate preferences.** RNA targets of the different TRAMP subunits were identified by in vivo UV-cross-linking followed by protein purification and sequence analysis of cDNAs (CRAC). We compared strains in which the endogenous gene was HTP tagged for Air1, Air2, Trf4, Trf5, and Mtr4, as well as the exosome exonucleases Rrp44 and Rrp6. The Mtr4 arch domain is implicated in substrate recruitment[13,14,38,39], so we also constructed and analyzed a tagged Mtr4 mutant lacking this region (Mtr4Δarch) (Supplementary Fig. 1A). The Mtr4Δarch construct was expressed from $P_{MTR4}$ in a strain in which wild-type (WT) Mtr4 was under $P_{GAL}$ control, allowing its depletion on glucose medium (Supplementary Fig. 1B).

Comparison of RNA target classes recovered with each protein showed clear differences (Fig. 2a). Notably, pre-rRNA spacer regions (external transcribed spacer (ETS) and internal transcribed spacer (ITS)) were substantially more targeted by Air1 than Air2, whereas Trf4 and Trf5 exhibited similar recovery. They were also less recovered with Mtr4Δarch relative to Mtr4, consistent with pre-rRNA maturation defects reported for strains carrying arch mutations (Supplementary Fig. 1C)[13,14,38,39]. In contrast, the major ncRNA classes, CUTs, SUTs, and XUTs, were strongly recovered with Air2 and Trf4, relative to Air1 plus Trf5. Notably, protein-coding genes appeared to be major targets of Trf5, which showed the strongest recovery compared to the other factors.

TRAMP targets are expected to be subject to oligoadenylation, so we specifically analyzed cDNAs that carry additional, 3′-terminal non-templated A residues (Fig. 2b). These are very likely to represent the authentic 3′ ends of RNAs that have been recognized by one of the TRAMP complexes and targeted for degradation by oligo(A) addition. In these datasets, the preference for Air1 over Air2 in pre-rRNA binding was even more marked. Similarly, the preference for Air2 and Trf4 in recovery of ncRNAs was more striking in the oligo(A) population. This was seen for CUTs, SUTs, and XUTs, as well as antisense and intergenic regions and RNAPIII transcripts ("other ncRNAs" in Fig. 2b).

Similarities and differences between TRAMP components were assessed by the degree of correlation between the recovered targets for RNAPII transcripts (Fig. 2c and Supplementary Fig. 2, Supplementary Data 4) As expected, target RNA species ("co-targeting") were more similar than the co-localization of the

precise binding sites for different factors ("co-localization"). Consistent with the proteomic analyses and target RNA classes, Air1 binding was similar to Trf4 and Trf5, while Trf4 showed similarity to both Air1 and Air2. In contrast, Trf5 binding was more similar to Air1 than to Air2, while Air2 more resembled Trf4 that Trf5. Notably, targets for Mtr4Δarch were substantially less well correlated with all other TRAMP components than intact Mtr4 (Fig. 2c). This might not have been anticipated, since the arch domain was implicated in targeting Mtr4 to degrade specific pre-rRNA spacer regions that are independent of TRAMP[15,16]. Higher correlations between different TRAMP components were observed for oligo(A) RNAs than considering all reads (Fig. 2c and Supplementary Fig. 2), confirming that filtering for adenylated species more reliably identifies bona fide surveillance targets.

**Trf5–Air1 recruits Mtr4 to the A2–A3 region of the pre-rRNA.** Mtr4 and the exosome are required for degradation of the excised 5′ ETS and the ITS2 region present in the 7S pre-rRNA. These are "default" activities, in that they are required during maturation of all pre-rRNAs, and neither is known to involve the TRAMP complexes. In both cases, Mtr4 is directly recruited to the pre-ribosome by binding to AIM-containing proteins, Utp18 and Nop53, respectively[16]. In contrast, the 23S RNA, an aberrant intermediate that extends from the transcription start to site A3 in ITS1, is also a characterized Mtr4 and exosome substrate, but is an subject to RNA oligoadenylation and surveillance[27]. To assess the roles of other TRAMP components in Mtr4 recruitment, we tested strains lacking Air1, Air2, Trf4, or Trf5. Notably, loss of the other TRAMP components did not affect Mtr4 abundance (Supplementary Fig. 1D).

Analysis of protein binding in ITS1 (Fig. 3 and Supplementary Fig. 3) using oligo(A) reads (Fig. 3) showed high binding of Air1, Trf5, and Mtr4 at positions 5′ to site A3 (the 3′ end of 23S) relatively to total reads (Supplementary Fig. 3A, D, E), whereas little association was seen for Air2 or Trf4 (Fig. 3b, c and Supplementary Fig. 3B, C). Mtr4 binding across ITS1 was also tested in strains deleted for TRAMP component (Fig. 3f–i and Supplementary Fig. 3F–I). Loss of Trf5 or Air1 strongly reduced Mtr4 association (~5.8- and ~3.7-fold, respectively) (Fig. 3f, i, m). The effects of air1Δ were more modest, probably due to its replacement by Air2 (see Fig. 1c). Strains lacking Air2 also showed a reduction in Mtr4 association with ITS1 (~3.2-fold decrease), while loss of Trf4 had a milder effect on Mtr4 binding at this site (~2.0-fold).

These observations probably reflect both alterations in the balance of the remaining TRAMP complexes, when one component is absent, and the significant redundancy between Air1/Air2 and Trf4/Trf5 proteins. In the absence of Air2, increased levels of free Trf4 are expected to compete with Trf5 for Air1 binding, leading to increased formation of TRAMP4-1

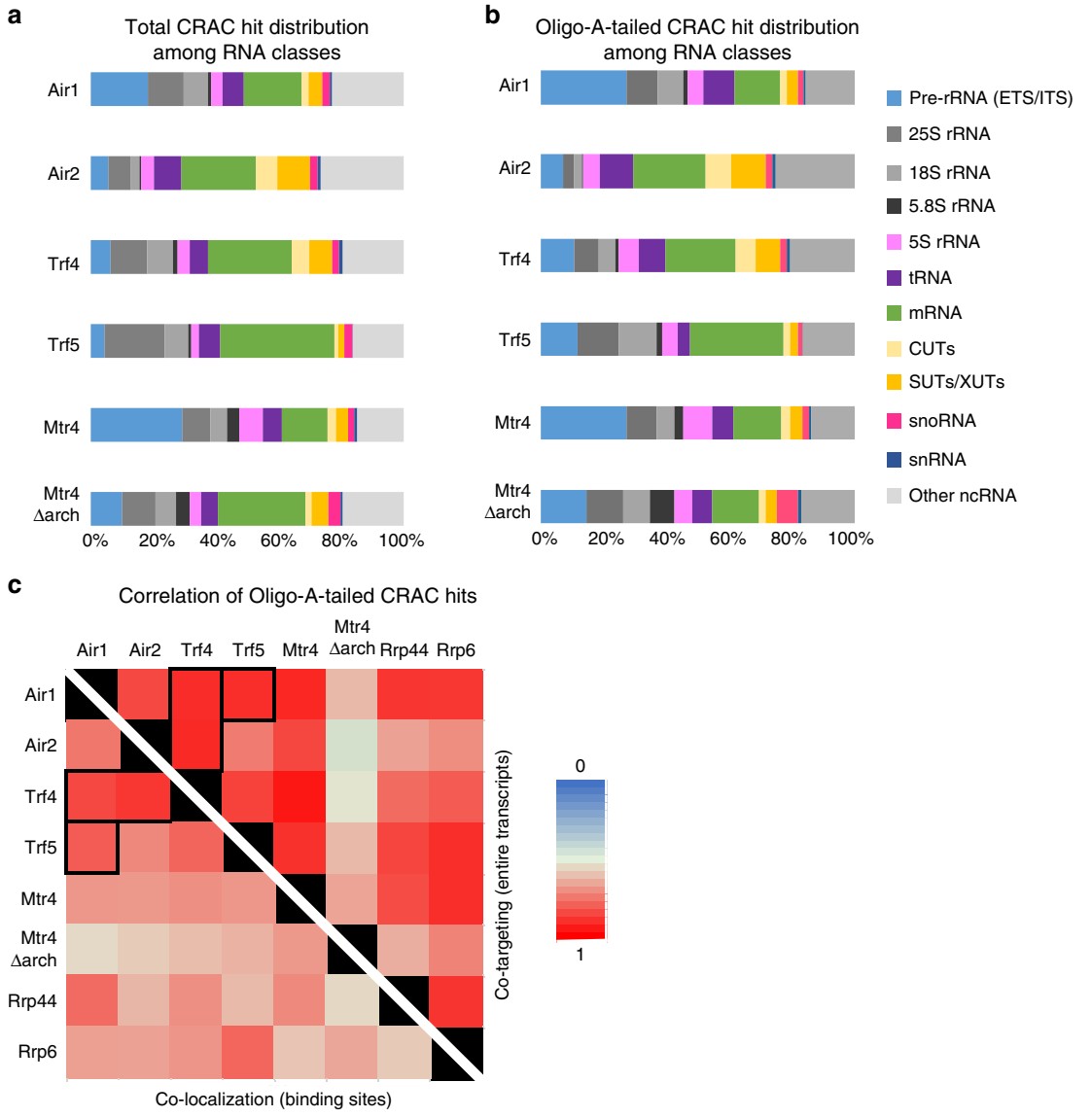

**Fig. 2 RNA interactions by TRAMP components. a**, **b** Distribution of all (**a**) and non-encoded A-tailed (**b**) reads mapped to different RNA substrate classes recovered in CRAC datasets. Two biological replicates were averaged for each protein. **c** Correlation matrix displayed as a heat map showing correlation between binding sites recovered with different factors. Note: Above the diagonal, the matrix shows the extent to which the factors target the same RNA species as "co-targeting of RNAs." Below the diagonal, the matrix shows the extent to which the factors target the closely positioned RNA sites (≥50 nt) as "co-localization of sites." See Supplementary Data 4 for values for individual CRAC dataset.

and correspondingly reduced TRAMP5-1. This will partially mimic effects of *air1Δ*, consistent with ITS1 binding specificity arising from Trf5.

To directly assess redundancy of the Air proteins and the altered balance of TRAMP complexes in strains deleted for TRAMP components, we performed CRAC on Air1 in an *air2Δ* strain, and on Air2 in *air1Δ* (Supplementary Fig. 4). In the *air2Δ* background, Air1 was less recovered (~2.5-fold decrease) across ITS1 relative to the *AIR2* strain (Supplementary Fig. 4A, C, E, G, I, J). We conclude that the absence of Air2 leads to its partial replacement by Air1, with increased TRAMP4-1 formation and correspondingly less TRAMP5-1 than in WT. This depletes binding across transcripts specifically targeted by Trf5. In contrast, binding of Air2 across ITS1 was minimally affected by loss of Air1 (Supplementary Fig. 4B, D, F, H–J), probably indicating that a Trf5–Mtr4–Air2 complex is not efficiently recruited to Trf5-specific targets. Similarly, in the absence of Trf4, TRAMP5-1 is expected to partially occupy previous TRAMP4-1

and 4-2 binding sites, reducing recovery at sites specifically occupied by Trf5 in the WT background.

Deletion of the arch domain from Mtr4 (Fig. 3j, m) also strongly reduced its association upstream of site A3. Similar profiles were observed for total, unfiltered reads (Fig. 3n and Supplementary Fig. 3E, J), confirming that the loss of Mtr4 binding in *trf5Δ* was due to a recruitment defect and not to reduced oligoadenylation. Neither Rrp44 nor Rrp6 showed clear binding across the A2–A3 region. We re-analyzed published data for exonuclease-deficient Rrp44 (Rrp44 exo), which is stabilized in binding to many exosome substrates, but this also did not show clearly elevated binding to A2–A3. Northern blot analysis showed strong accumulation of cleaved A2–A3 fragment in *trf5Δ* compared to WT, confirming the role of Trf5 in degradation of this RNA (Fig. 3o). No comparable effect was observed for *trf4Δ*.

We conclude that recruitment of Mtr4 to the A3-cleaved pre-rRNA requires both Trf5 and the arch domain (see Fig. 3p for a cartoon). The finding that the loss of Mtr4 was modest in the

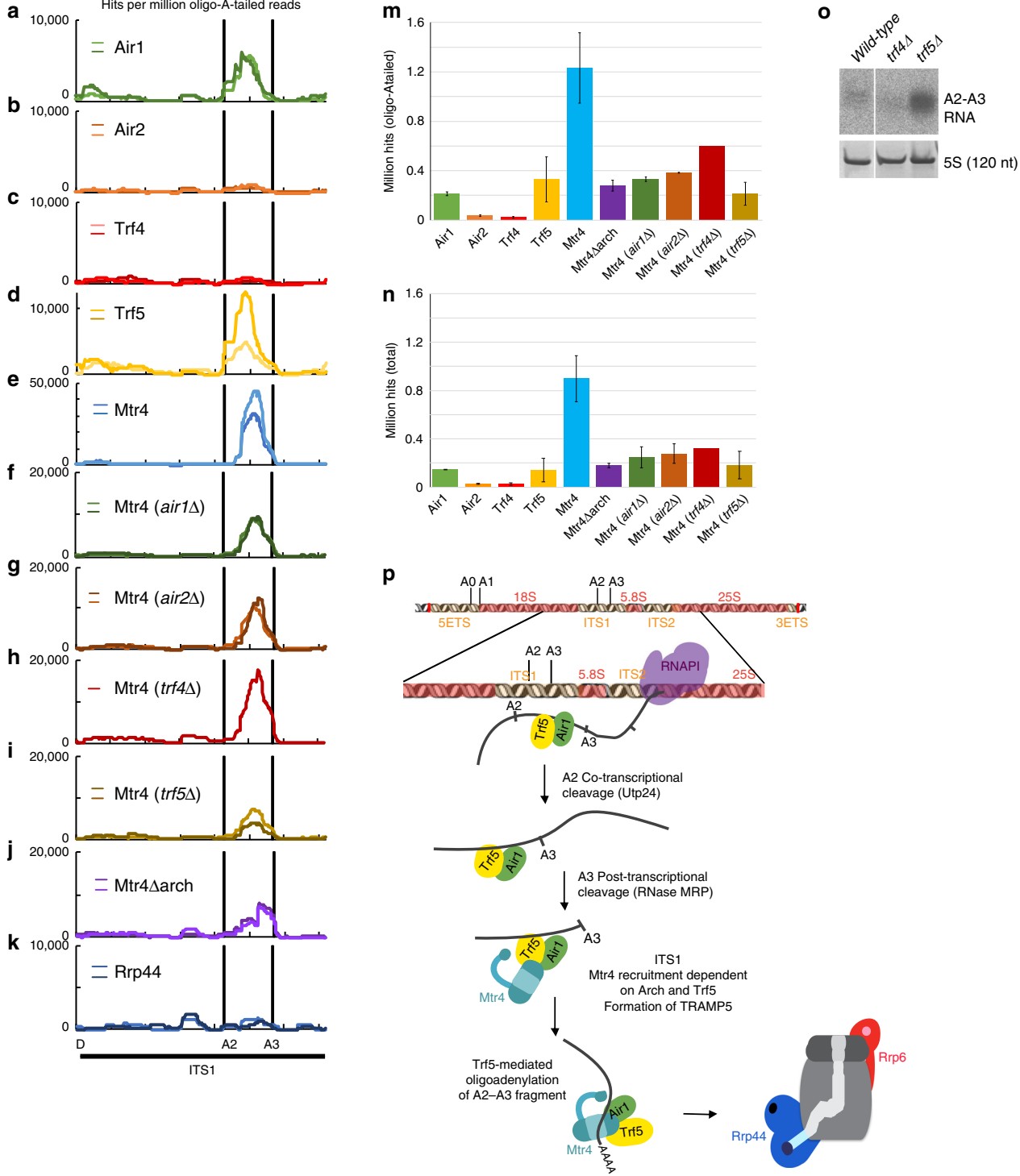

**Fig. 3 TRAMP interaction sites on the pre-rRNA ITS1 region. a–k** Distribution of non-encoded oligo(A)-tailed reads, normalized to millions of total mapped reads, across the ITS1 region of the pre-rRNA. The graph shows the distribution of reads recovered with the indicated TRAMP components (**a–e**), Mtr4 in strains lacking the indicated TRAMP components (**f–l**), Mtr4 lacking the arch domain (**j**), and Rrp44 (**k**). Scale is linear. A diagram of the pre-rRNA region is also shown. Two independent replicates are shown in panels as light and dark colored lines. **m, n** Hits per millions oligo(A)-tailed reads (**m**) and total reads (**n**) encompassing the A2–A3 region of ITS1 extended by 10 nt from each side were summed and are exhibited as a bar diagram. Two individual replicates were averaged, with standard deviation shown as error bars (except for Mtr4 (*trf4Δ*) for which only one experiment was performed). **o** Northern blot analysis with probe for the A2–A3 fragment (probe for 5S rRNA was used as a loading control). **p** Model for surveillance within ITS1. Trf5 and Air1 are recruited co-transcriptionally to the A2–A3 region. Following A2 cleavage, ribosome synthesis factors are released, freeing site A3. Mtr4 is recruited, probably via protein–protein interactions involving Trf5 and an Mtr4 arch interaction. Following A3 cleavage, Trf5 oligo-adenylates the free 3′ end prior to degradation by the exosome.

absence of Air1 indicated that this is largely redundant with Air2. This was unexpected, since it had long been envisaged that RNA-binding specificity would largely be determined by Air1 and Air2, which are Zn-knuckle RNA-binding proteins[4,5,26,40].

In an attempt to identify regions of Trf4 or Trf5 responsible for target specificity, we generated Trf5 constructs with non-conserved domains deleted or exchanged with Trf4. Unfortunately, neither deletion nor exchange of the N-terminal domain resulted in stable protein accumulation. Stable fusion constructs were expressed in which the C-terminal domain was deleted (Trf5ΔCTD) or replaced by Trf4 C-terminal domain (Trf5-CTD4), or in which a 5-aa, AIM-like sequence was replaced by 5 Ala (Trf5-5xA). In CRAC analyses, these constructs all showed inefficient cross-linking, resulting in poor signal-to-noise ratios and variable results (Supplementary Fig. 6C–G and Fig. 8). However, the Trf5 C-terminal domain appeared to be important for recruitment of Trf5 to pre-rRNA (Supplementary Fig. 6C), particularly across the A2–A3 region of ITS1 (Supplementary Fig. 6E, F), whereas the AIM-like domain appeared dispensable for recruitment (Supplementary Fig. 6G).

**Trf4 and Trf5 are enriched on different regions of pre-mRNAs.** The distribution of TRAMP components on protein-coding genes was initially assessed in metagene analyses of mRNAs aligned by the transcription start sites (TSS) or pA sites (Fig. 4a–m; all protein-coding genes >500 nt in length were included in metagene plots).

Strong, promoter-proximal enrichment of the exosome has previously been reported[41,42], reflecting degradation of ncRNAs generated by early RNAPII termination (shown for the exosome catalytic subunits Rrp6 and Rrp44 in Fig. 4f, l). Similar 5′ enrichment was seen for Mtr4 (Fig. 4a), Air2 (Fig. 4c), and Trf4 (Fig. 4d), as well as the exosome-associated nucleases Rrp6 and Rrp44 (Fig. 4l–n). In marked contrast, pA site-proximal peaks were seen for Trf5, Rrp6, and to a lesser extent for Air1, but not for Trf4, Air2, or Rrp44. In addition, a modest pA-proximal peak was seen for Mtr4.

Analysis of mRNA binding by Air1 in air2Δ and Air2 in air1Δ strains (Supplementary Fig. 4K–N) revealed profiles that are much more similar than in the WT background. In the absence of Air2, recovery of Air1 was elevated on mRNAs, particularly in the TSS-proximal region where Air2 is normally bound. In the absence of Air1, recovery of Air2 was elevated at the pA site, where a peak of Air1 is otherwise observed. These observations strongly support the model that the specificity of binding by TRAMP complexes is largely conferred by Trf4 and Trf5.

All TRAMP components showed substantial numbers of hits within pre-mRNA introns (Supplementary Fig. 5), with TRAMP4-1 apparently most strongly targeting introns. Notably, the arch domain appears to be dispensable for Mtr4 recruitment to introns.

The 5′ peak for Mtr4 was reduced by either deletion of the arch (Fig. 4g) or the absence of Trf4 (Fig. 4j), but was not clearly altered by loss of Trf5, Air1, or Air2. Recruitment of Trf4 to the TSS was not clearly affected by Mtr4Δarch (Fig. 4f and Supplementary Fig. 1E). These results are consistent with arch-dependent recruitment of Mtr4, with specificity dependent on Trf4. Surprisingly, recruitment of Rrp44 was unaffected in Mtr4Δarch (Fig. 4m, n). The same analysis was performed considering only the mRNAs for which Mtr4 binding was most dependent on the arch domain; that is, the 200 mRNAs that showed the greatest reduction in recovery with Mtr4Δarch compared to Mtr4 (Fig. 4o, p). On these mRNA, the TSS-proximal peak of Rrp44 was substantially decreased, indicating that the Mtr4 arch domain plays a role in TSS-proximal

recruitment of the exosome on specific targets (see model in Fig. 4q).

**Distribution of TRAMP components on individual mRNAs.** The metagene plots indicated differential recruitment of Trf4 and Trf5, so we next assessed whether this reflected preferential association with different mRNA subsets. The top 1000 mRNAs in the CRAC datasets for each TRAMP component were combined (2005 mRNAs in total). All mRNAs were divided into five bins of equal length and the distribution of each factor was determined for every bin. These data were used to cluster genes with related patterns of factor distribution (Fig. 5a and Supplementary Fig. 7, Supplementary Data 5). This analysis identified groups of mRNAs with distinctly different TRAMP occupancy.

The largest group of mRNAs was cluster 3 (Fig. 5a and Supplementary Fig. 7C, I), which showed high TSS-proximal occupancy of Trf4, Air2, and Mtr4, with somewhat lower binding for Air1 and Trf5. This is consistent with the metagene profiles in Fig. 4. In contrast, cluster 2 showed high poly(A) proximal occupancy of Trf5, Air1, and Mtr4, with lower total binding of Air2 and Trf4 (Fig. 5a and Supplementary Fig. 7B, H). These features are also seen in metagene plots (Fig. 4). Other clusters showed distinct features: in cluster 1 (Fig. 5a and Supplementary Fig. 7A, G), Trf5 showed substantial 3′ occupancy, along with a mild enrichment for Air1 (Supplementary Fig. 8). Cluster 1 mRNAs are very likely targeted by two distinct pathways: a minor one in which TRAMP4-1 plus TRAMP4-2 bind proximal to the TSS and the major pathway in which Trf5 targets the 3′ end, probably partly with Air1. In consequence, Air1 hits within the mRNA are split between TSS and the pA site. Since the data are normalized across each mRNAs, this results in a reduced 3′ peak for Air1 relative to Trf5 (Supplementary Fig. 8). Rrp6 and Rrp44 are not clearly 3′ enriched on these mRNAs, suggesting that their interaction with Trf5 may not be linked to surveillance. In cluster 4, Trf4, Trf5, Air1, Air2, Rrp6, and Rrp44 all show moderate TSS-proximal binding, but Mtr4 is largely absent. Surprisingly, TSS-proximal binding was relatively increased for Mtr4 in trf4Δ or trf5Δ strains and for Mtr4Δarch (Fig. 5a and Supplementary Fig. 7D). However, total association for Mtr4 was reduced in the absence of Trf4 (Supplementary Fig. 7J).

The identified clusters in the heat map were analyzed for GO term enrichment of the protein products, as an indication of functional classes[43] (Fig. 5b). Each cluster showed highly significant enrichment for mRNAs encoding specific, functional subsets of proteins. This provides strong evidence that the TRAMP-binding patterns observed have functional consequences for the regulation of protein production.

**Correlations in pre-mRNA-binding sites.** Trf4 and Trf5 appear to recruit Mtr4 to at least some nuclear pre-mRNA sites, but the origin of their binding specificity was unclear. As an initial attempt to address this, we compared the distribution of TRAMP components to other nuclear RNA-binding proteins that have been similarly mapped using CRAC (Fig. 6 and Supplementary Fig. 2, Supplementary Data 4 and 6). For this we compared both the total relative recovery over each RNA species or the extent of binding at closely positioned RNA sites (closer than 50 nt) (designated as "co-targeting of RNAs" and "co-localization of sites," respectively, in Fig. 6a and Supplementary Fig. 2). These analyses were performed either genome wide across all annotated genes (Supplementary Fig. 2, Supplementary Data 4) or over all annotated RNAPII transcribed genes (Fig. 6a and Supplementary Data 6).

Some expected results were obtained; the nuclear poly(A) binding factors Pab1 and Nab2 were closely correlated and

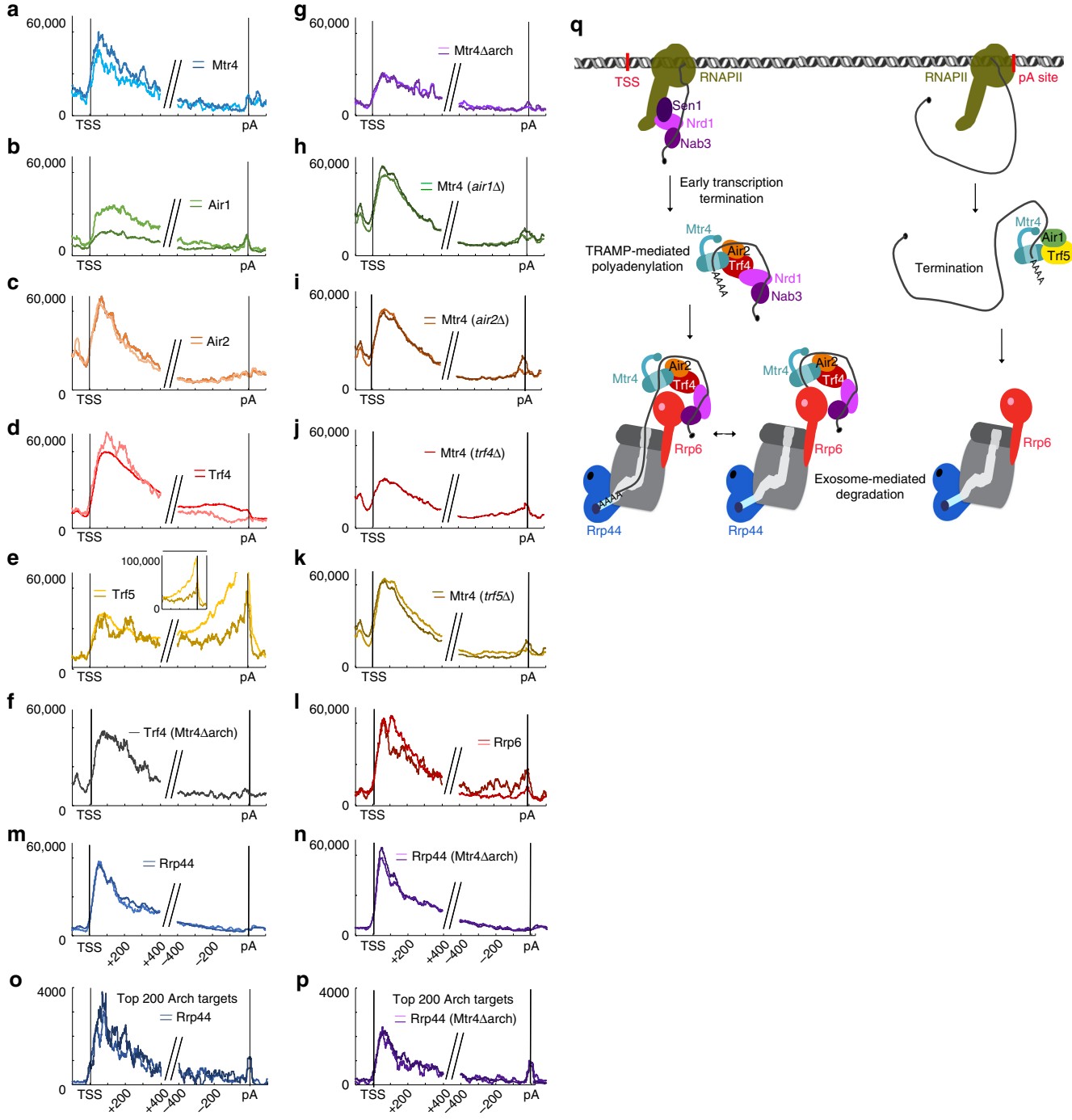

**Fig. 4 Metagene analyses of TRAMP components across protein-coding genes.** Binding by RNA surveillance and degradation factors is strongly enriched close to the TSS on protein-coding genes. **a**–**n** Distribution of individual components indicated, across all mRNAs longer than 500 nt. Each panel shows the hit density, normalized to millions reads mapped to mRNAs. The two lines in each panel represent results from independent CRAC experiments. Reads were aligned with transcription start sites (TSS) and polyadenylation sites (pA). **o**–**p** Distribution of Rrp44 in strains expressing Mtr4 or Mtr4Δarch across the 200 mRNAs showing the greatest dependence on the arch for Mtr4 recruitment. Each panel shows the hit density, normalized to millions of reads mapped to mRNA, for all genes longer than 500 nt. The two lines in each panel represent results from independent CRAC experiments. Reads were aligned with transcription start sites (TSS) and polyadenylation sites (pA). **q** Model representing proposed recruitment pathways. Close to the TSS, Nrd1–Nab3–Sen1-mediated early transcription generate TSS-proximal ncRNAs and recruits TRAMP4-2 through Trf4 and interactions involving the arch domain. Following Trf4-mediated oligoadenylation, RNAs are targeted to both Rrp44 and Rrp6 nucleases activity of the exosome. A distinct surveillance mechanism occurs at some pA sites involving specifically TRAMP5-1 and Rrp6, which can also be directly recruited by Nab3[63].

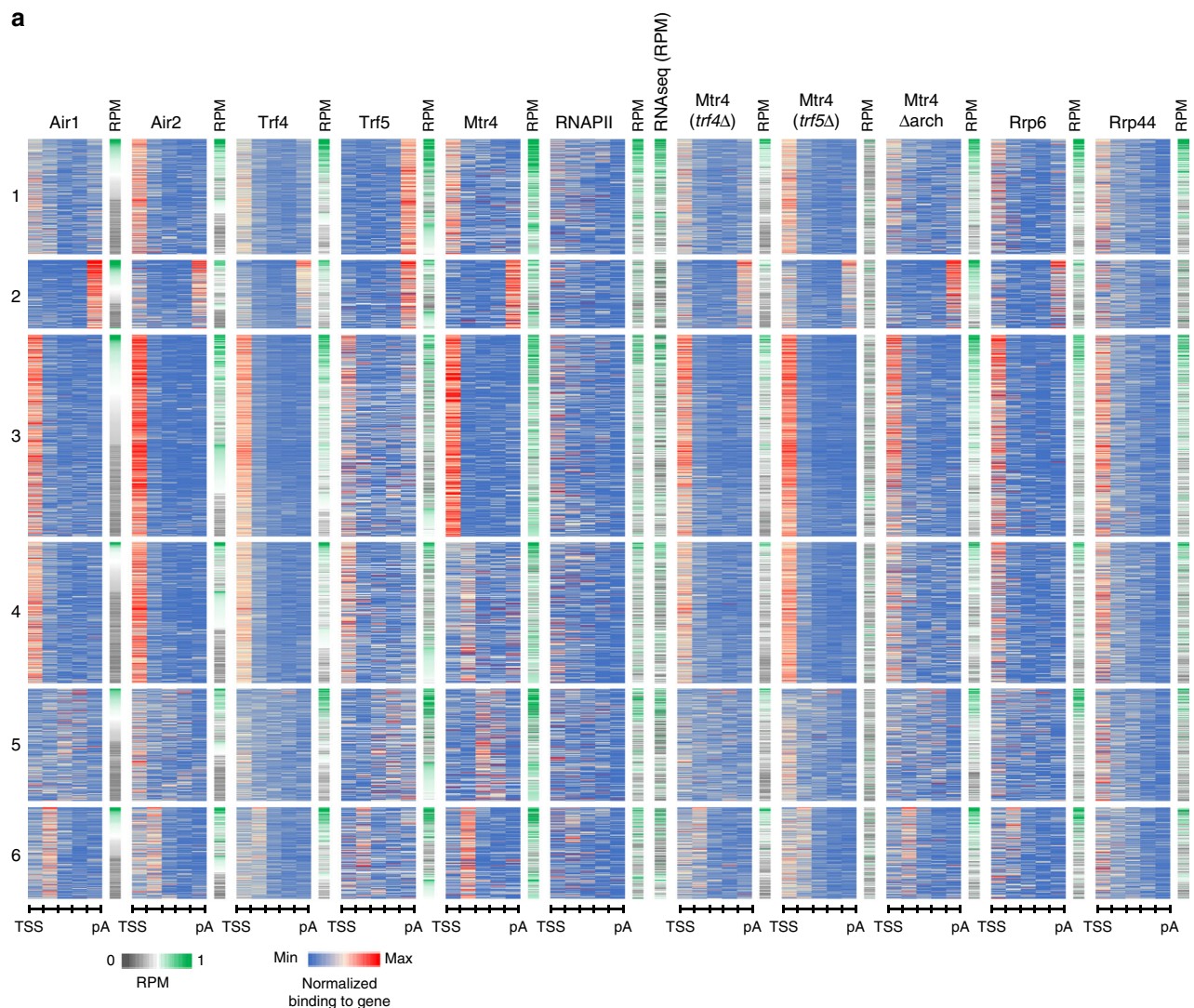

**Fig. 5 Heat maps for distribution of TRAMP components on each mRNA. a** Heat map showing binding across individual mRNAs of different factors. The top 1000 protein-coding genes recovered in CRAC with each TRAMP component were selected and combined (2005 mRNAs in total). Each transcript was divided into five bins of equal length from TSS to pA site. Binding across each bin was calculated as a fraction of total binding across individual gene (set to 1). The number of reads was averaged between two biological replicates. Only the coverage of Air1, Air2, Trf4, Trf5, and Mtr4 along each transcript was used for Euclidean distance-based clustering. Other datasets shown were not included in the clustering analysis but are sorted according to the clustered list. The data are displayed as heat maps. The RPM for each transcript recovered by each protein was calculated and represented as independent heat maps. See Supplementary Data 5 for cluster composition. **b** Enriched GO terms[43] and features for clusters of mRNA defined in **a**.

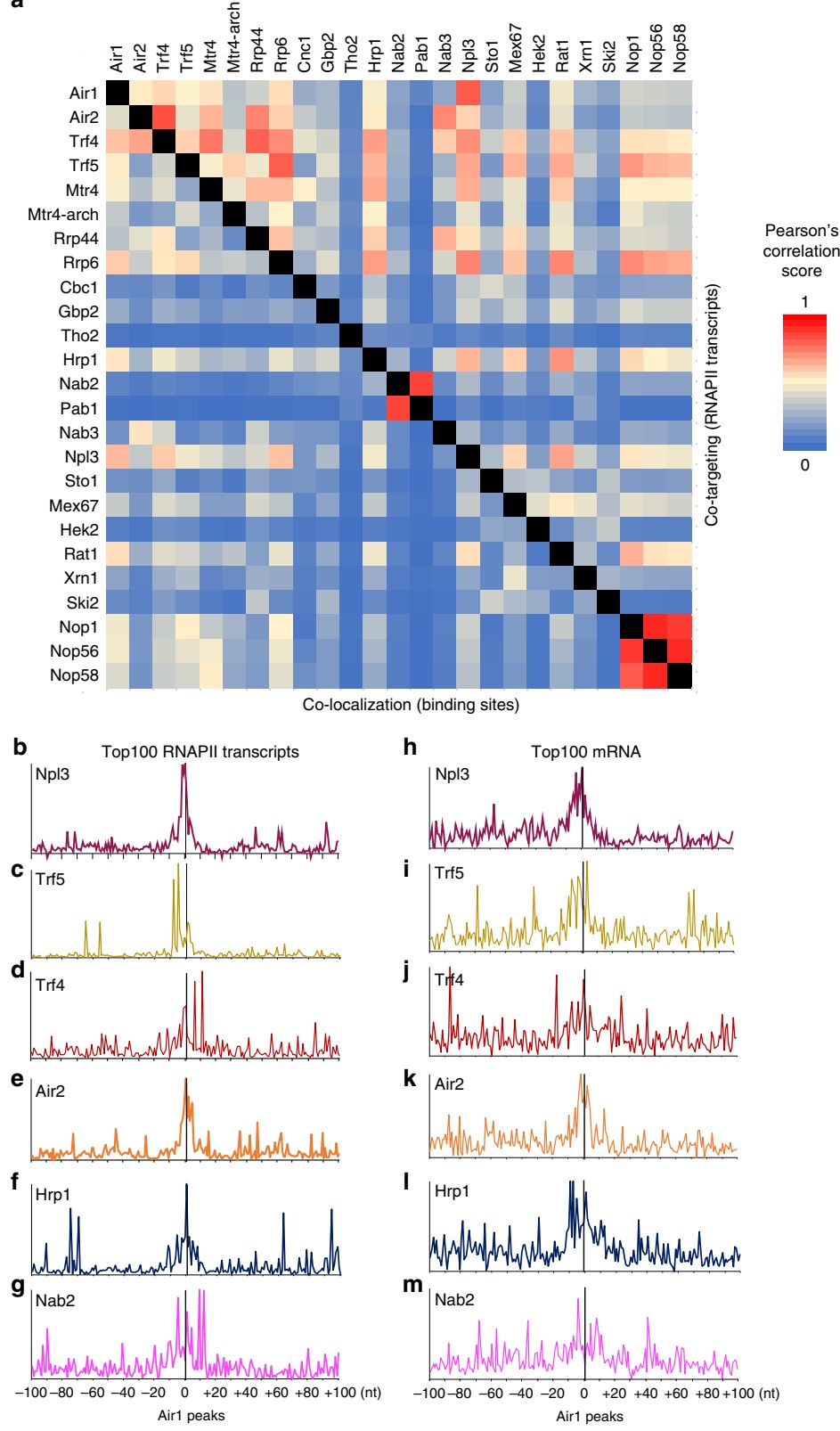

**Fig. 6 Correlations between protein interaction sites on RNAPII transcripts. a** Correlation matrix displayed as a heat map, showing correlations between binding sites recovered with different factors across RNA polymerase II transcripts. Note: Above the diagonal, the matrix shows the extent to which the factors target the same RNA species as "co-targeting of RNAs." Below the diagonal, the matrix shows the extent to which the factors bind closely positioned RNA sites (≥50 nt) as "co-localization of sites." **b–m** Binding of the indicated proteins relative to Air1. CRAC peaks for Air1 across the genome were selected and used as a reference to align peaks for other proteins. Data are shown for the combined top 100 RNAPII transcripts bound by Npl3 and Air1 (**b–g**, 127 genes in total) or top 100 mRNAs (**h–m**, 147 genes in total). Two independent replicates were combined in each panel.

distinct from most other factors. However, Nab2, which has been implicated in RNA surveillance[41], showed closer correlation with other surveillance factors (Fig. 6a). Similarly, the snoRNP proteins Nop1, Nop56, and Nop58 were closely correlated over RNAPII transcripts (Fig. 6a) or all genes (Supplementary Fig. 2). Ribosome synthesis factors (Utp proteins) were closely correlated on all genes, and showed correlations with surveillance factors, reflecting their interactions on pre-rRNAs.

For TRAMP components, Trf5 was better correlated with Air1 than with Trf4 or Air2, while Trf4 and Air2 were most highly correlated, consistent with the analyses of individual genes. The TRAMP components were also correlated with the exosome, as expected, although it was notable that Rrp6 was more correlated with Trf5, while Rrp44 correlated better with Trf4. Nab3 showed higher correlation with Air2 and Trf4 than Air1 and Trf5, consistent with Trf4 being recruited by Nrd1–Nab3 on some targets, including TSS-proximal ncRNA transcripts.

Hrp1, which was implicated in RNA surveillance[41], also showed correlations with TRAMP factors. A number of other proteins, showed notably close correlations, including the pre-mRNA-binding protein Npl3 with Air1. It may be relevant that Air1 was initially identified through an interaction with Npl3[9]. To follow-up this observation, Air1 CRAC peaks across the genome were selected using a peak-calling algorithm and used as a reference point to align reads from pre-mRNA-binding proteins and TRAMP components (Fig. 6b–m and Supplementary Fig. 10). Binding sites were compared across the top 100 bound RNAPII transcripts, which includes the abundant snoRNA species (Fig. 6b–g), or considering only mRNAs (Fig. 6h–m). Close co-localization was seen for Air1 and Air2 (Fig. 6e, k), supporting the conclusion that they are functionally redundant on many substrates. Consistent with the data in Fig. 6a, the peak of Air1 binding sites closely correlated with a peak of Npl3 binding over the top RNAPII transcripts (Fig. 6b; these include many snoRNAs) or mRNAs (Fig. 6f). Co-localization with Air1 was also seen for the surveillance factor Hrp1 (Fig. 6f, l).

Similar analysis were performed on Trf5ΔCTD, Trf5-CTD4, and Trf5-5xA (Supplementary Fig. 9). As for the pre-rRNA analyses (Supplementary Fig. 6), these constructs gave poor cross-linking on mRNAs. However, mutation or deletion of the CTD increased Trf5 association with the TSS region (Supplementary Fig. 9A). Notably, deletion of the CTD apparently lead to more delocalized binding across almost all mRNAs (Supplementary Fig. 9C), presumably reflecting a loss of specificity.

**Loss of Trf5 alters Mtr4 association and target mRNA abundance.** Previous functional and structural analyses of TRAMP have largely focused on Trf4 and Air2[21–25]. However, the heat maps (Fig. 5) indicated that subsets of nuclear mRNAs show preferential enrichment for binding to Trf5 (clusters 1 and 2). We therefore determined whether Trf5 has a functional impact on Mtr4 recruitment and mRNA stability.

mRNAs were stratified by ranking in Trf5 association, based on reads per kilobase per million mapped reads (RPKMs) across the genome. Bins of 200 mRNAs were compared between *TRF5* and *trf5Δ* backgrounds for changes in Mtr4 association by CRAC (Fig. 7a) or RNA abundance by RNA-sequencing (RNAseq) (Fig. 7b and Supplementary Fig. 11). Strikingly, mRNAs that were most strongly bound by Trf5 showed greatest reduction in Mtr4 binding when Trf5 is absent. Note that these are normalized data, so total recovery of Mtr4 across all genes is constant. The effects were quite substantial, with nearly 4-fold reduced Mtr4 association over the genes that were most strongly bound by Trf5. Conversely, mRNAs with the strongest Trf5 binding were modestly increased in the absence of Trf5, again using normalized

sequencing data. These data suggest that loss of Trf5 reduces Mtr4 recruitment with a consequent increase in pre-mRNA or mRNA stability. Changes in abundance shown in the RNAseq data were confirmed by real-time quantitative PCR on selected mRNAs (Supplementary Fig. 11A, Supplementary Table 7) relative to the ncRNA SCR1. The trend for increased mRNA abundance in *trf5Δ* was conserved between RNAseq and quantitative PCR (qPCR). However, greater increases were measured in qPCR, probably reflecting differences in normalization, which is complicated in surveillance mutants that potentially affect all RNA species.

A notable finding from the heat maps was the identification of group of mRNAs showing strong 3′ association with Trf5 and Air2 = 1 in the absence of Mtr4 or the exosome (cluster 1 in Fig. 5 and Supplementary Fig. 8) suggesting a distinct function for TRAMP5-1 here. To assess potential roles of Trf5, we assessed the distribution of RNAseq reads across mRNAs in WT and trf5Δ strains (Fig. 7c, d). For cluster 1, comparison of the WT and *trf5Δ* strains showed a clear deficit in reads close to the poly(A) site in the absence of Trf5 (Fig. 7c). Strikingly, comparable effects were not seen for any other cluster. This is presented for cluster 2, which also showed a 3′ peak of Trf5, but accompanied by Trf4, Mtr4, and Rrp6, and cluster 3, which showed only a 5′ peak of Trf5 and other TRAMP components (Fig. 7c). When mRNAs were ranked by Trf5 binding in bin 5 (the 3′ end), the same trend was seen (Fig. 7d). However, even for the most highly bound group, the 3′ depletion in mRNAs reads was less marked than for cluster 1.

We conclude that mRNAs bound by Trf5 are selectively affected by its absence. Notably, the clustering in Fig. 5 was entirely orthogonal to the mRNA sequence data in Fig. 7, indicating that the specific 3′ binding by Trf5 identified in cluster 1 is robustly correlated with stabilization of this region of the transcripts.

## Discussion

To better understand the targeting of RNA degradation and surveillance targets in vivo, we characterized the protein composition of the TRAMP complexes and identified specific binding sites for the different TRAMP components in WT and mutant cells. These analyses identified three complexes containing Mtr4 together with Trf4 and Air1 (TRAMP4-1), Trf4 and Air2 (TRAMP4-2), or Trf5 and Air1 (TRAMP5-1). Substantial differences in RNA binding were observed, indicating that TRAMP4-1, TRAMP4-2, and TRAMP5-1 each exhibit substrate specificities.

We note that, based on 14 different publications, *Saccharomyces cerevisiae* genome (SGD) (www.yeastgenome.org) lists median abundances (copies per cell) for TRAMP components as: Air1 (1851 ± 466); Air2: (1750 ± 420); Trf4 (2659 ± 411); Trf5 (1329 ± 608). If all components are in TRAMP complexes, this suggests very approximate abundances of: TRAMP4-1 (600); TRAMP4-2 (1700); TRAMP5-1 (1300). These values are consistent with the conclusion that Trf4 interacts with Air1 and Air2, whereas Trf5 binds only Air1.

TRAMP5-1 preferentially targeted the ITS1 spacer region of 35S pre-rRNA, a characterized exosome substrate for which no AIM domain ribosome synthesis factor has been identified. Mtr4 binding to ITS1 in the pre-rRNA was strongly reduced by loss of Trf5, whereas loss of Trf4, Air1, or Air2 had only modest effects. In contrast, TRAMP4-2 was more strongly associated with RNAPII transcripts, particularly mRNA 5′ ends, close to the TSS and with the CUT, SUT, and XUT ncRNAs. This implicated TRAMP4-2 in degradation of ncRNAs, including promoter-proximal ncRNAs generated by early termination of transcription,

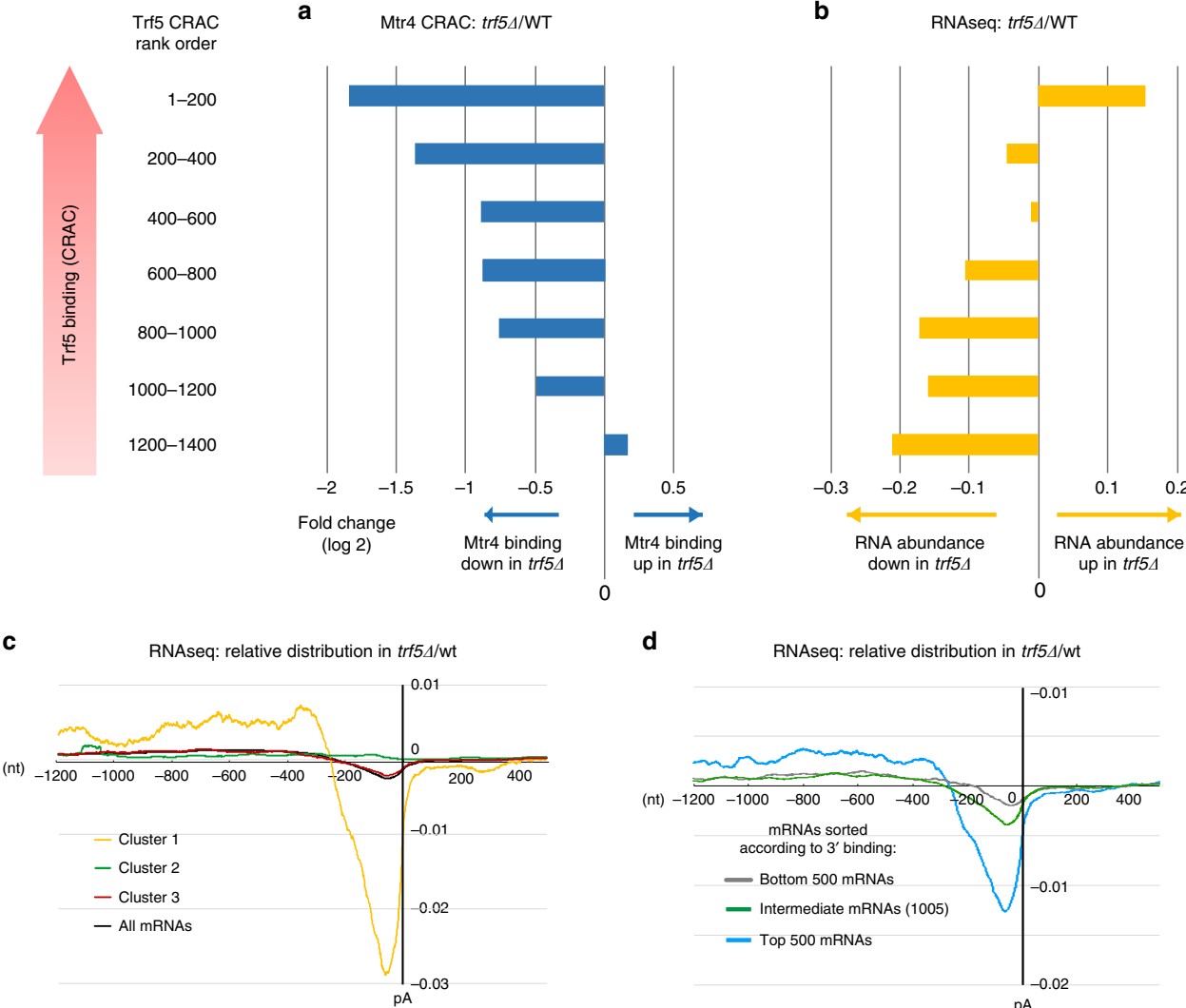

**Fig. 7 Alterations in mRNA abundance in strains lacking Trf5. a** Changes in Mtr4 binding by CRAC in strains lacking Trf5, relative to the wild type. **b** Changes in mRNA abundance by RNAseq in strains lacking Trf5, relative to the wild type. Genes are ordered by ranking in frequency of Trf5 binding (RPKM) and bars show fold change (log 2) in strains lacking Trf5 relative to the wild type. Note that these are normalized data, so the total recovery of Mtr4 across all genes is constant. **c, d** RNAseq distribution across mRNAs, normalized to the total binding value across the mRNA subset, in wild-type (four independent replicates averaged) and *trf5Δ* (three independent replicates averaged). Reads were aligned with the polyadenylation sites (pA). **c** shows genes grouped according to clusters identified in Fig. 5a, plus all mRNAs. In **d**, the 2005 genes analyzed in Fig. 5 are grouped by Trf5 binding in the 3′ region (bin 5 in Fig. 5a and Supplementary Fig. 7).

a model supported by co-localization with Rrp44 and Rrp6. Mtr4 exhibited lower binding to TSS-proximal regions when Trf4 was absent, with comparable reduction on deletion of the arch domain, but was little altered by loss of Air1 or Air2. In the absence of its paralog, the binding profiles of Air1 and Air2 across mRNAs are highly similar, strongly indicating that they are not directly responsible for the substrate specificity seen in the WT background. We conclude that substrate specificity is predominantly provided by Trf4 and Trf5, while Air1 and Air2 appear largely interchangeable. This was unexpected, since it had previously been anticipated that RNA-binding specificity would largely be determined by Air1 and Air2, which are Zn-knuckle RNA BPs[26,40], or by specific interactions of AIM-containing proteins with the arch domain of Mtr4[15,16].

When bound to the exosome, Mtr4 is positioned at the entrance to the central lumen, such that RNAs can pass through the helicase domain directly into the exosome[44,45]. The structure of the Trf4–Air2–Mtr4 TRAMP complex shows the arch located opposite Trf4–Air2[22]. N-terminal, low-complexity regions of Air2 and Trf4 are jointly responsible for binding to the DExH core of Mtr4 in vitro. The residues in Trf4 and Air2 that bind the Mtr4 helicase core, are conserved in Trf5 and Air1, respectively. The N terminus of Trf4 (aa 1–110) is absent from the TRAMP structure, but could conceivably extend to the arch domain of Mtr4. However, a more likely interaction is between Air2 and the arch. Indeed, in the crystal structure, the N terminus from the adjacent Air2 molecule in the crystal lattice was shown to contact the Mtr4 arch at the same sites as the AIM of Nop53[15,22], through residues that are conserved in Air1. Association of Air2 and Air1 with the arch domain might require conformational changes in which the arch moves toward the helicase core, switching Mtr4 (and the TRAMP) to a closed conformation. We therefore postulate that recruitment of Mtr4 to most nuclear RNA surveillance targets is based on the specificity of Trf4 or Trf5 interactions with the target, followed by (largely redundant) binding of Air1 or Air2 to the arch domain.

Comparison of TRAMP binding with other factors expected to interact with nuclear transcripts showed a range of concordances. We speculate that combinations of proteins binding to nascent transcripts act together to promote or disfavor binding by the surveillance machinery and rapid nuclear RNA degradation. Particularly notable was the correlation between the hnRNP-like, pre-mRNA binding protein Npl3 and Air1, which was initially isolated via interaction with Npl3[9]. Air1–Npl3 binding is bridged by the arginine-methyl transferase Hmt1 and blocks arginine methylation on Npl3, potentially changing its functional properties.

Clustering of the most recovered ~2000 mRNAs, based on TRAMP factor occupancy, identified six clusters. Five of these showed statistically significant enrichment for functional classes of protein products, while the remaining class was enriched for types of transcript; introns with encoded features, notably snoRNAs; and unconfirmed protein products. Such functional enrichment strongly supports the significance of TRAMP factor binding in regulating gene expression.

A notable finding was the apparently specific binding of Trf5 to a cluster of mRNAs. To better understand the significance of this association, we assessed the effects of deletion of *TRF5* on Mtr4 binding by CRAC and RNA abundance by RNAseq. When mRNAs were stratified by Trf5 association, mRNA species most highly bound by Trf5 in the WT, showed decreased binding by Mtr4 and increased total abundance in the *trf5Δ* strain. This is consistent with Trf5 promoting Mtr4 recruitment and degradation for many pre-mRNAs.

We also noted that clusters of mRNAs showed high p(A)-proximal association of Trf5 (clusters 1 and 2 in Fig. 5a). In particular, the 3′ association of Trf5 and Air1 with mRNAs in cluster 1 was not accompanied by clear association with Mtr4 or the exosome, suggesting a possible function distinct from RNA surveillance and degradation. Comparing RNAseq data across mRNAs revealed a deficit in 3′reads, specifically for cluster 1. No such deficit was observed for cluster 2, in which 3′ accumulation of Trf5 is accompanied by binding of Mtr4 and other TRAMP components. It remains unclear whether this apparent truncation reflects altered pre-mRNA synthesis or stability, but Trf5 clearly has significant effects on the 3′ ends of highly bound pre-mRNAs.

We conclude that the functions of the different yeast TRAMP complexes have significantly diverged, particularly for the nucleotide transferases Trf4 and Trf5.

## Methods

**Strains**. All yeast strains are derived from (BY4741, *MATa; his3Δ1; leu2Δ0; met15Δ0; ura3Δ0*). Standard techniques were used to integrate C-terminal affinity tags and integration of galactose-driven genes. Strains used are listed in Supplementary Data 1.

For Mtr4 depletion, strains were grown in 2% galactose minimal media, then shifted to 2% glucose media for 16 to 20 h until exponential phase (optical density (OD) ~0.5) and used for CRAC or RNA extraction. Depletion was checked by western blotting.

**Western blotting**. Yeast was grown to exponential phase (OD 0.5). Proteins were extracted with a mild alkali treatment prior boiling in NuPaGE loading buffer[46]. Yeast from 4OD of cells were loaded on NuPaGE Novex 4–12% gel, liquid transferred and incubated with rabbit anti-TAP antibody (Thermo Scientific CAB1001, 1:5000) rabbit anti-TAP antibody (Thermo Scientific CAB1001 and mouse anti-Pgk1 antibody (Thermo Fisher PA528612; 1:5000) and mouse anti-Pgk1 antibody (Thermo Fisher PA528612), followed by IRDye secondary antibodies (Licor) incubation (dilution 1:10,000): anti-mouse 680RD (926-68070) and anti-rabbit 800CW (926-32211) incubation. Membrane was visualized on Odyssey CLx scanner.

**Northern blotting**. RNAs were extracted via a hot phenol, guanidium method[47]. Oligoadenylated RNAs were purified from 75 μg of total RNAs using polyA+ kit (Ambion). All outputs, along with 5 μg total RNA, were separated on a 10% acrylamide urea gel, stained with SybrSafe (Invitrogen) and liquid transferred to a HybondN+ membrane. Hybridization with a radiolabeled oligonucleotide (GCGTTGTTCATCGATGC) was performed with ultraHyb (Ambion) and signal detected using a Fuji FLA-5100 PhosphoImager.

**Protein sequence alignment**. Alignment were generated with Clustal Omega and visualized using MView 1.63[48].

**MS analyses**. Protein pull-downs were extracted from strains expressing the bait protein tagged with a C-terminal HisX6-Tev cleavage site-Protein A (HTP) tag. A non-tagged strain was used as a negative control. Protein purification has been made in 1 M NaCl conditions. For Trf4-HTP and Trf5-HTP, additional purification with lower stringency (150 mM NaCl) were performed in the presence or absence of RNase A. Extracts were separated by sodium dodecyl sulfate-polyacrylamide gel electrophoresis, trypsin-digested, as previously described[49]. Following digestion, samples were diluted with equal volume of 0.1% trifluoroacetic acid and spun onto StageTips as described[50]. Peptides were eluted and analyzed by liquid chromatography-tandem MS on a Q Exactive Mass Spectrometer (Thermo Fisher Scientific) coupled online, to an Ultimate 3000 RSLCnano Systems (Dionex, Thermo Fisher Scientific). Fourier transform mass spectrometry spectra were recorded at 70,000 resolution and the top 10 most abundant peaks with charge ≥2 and isolation window of 2.0 Thomson were selected and fragmented by higher-energy collisional dissociation[51] with normalized collision energy of 27. The maximum ion injection time for the MS and MS2 scans was set to 20 and 60 ms, respectively, and the automatic gain control target was set to one E6 for the MS scan and to five E4 for the MS2 scan.

The MaxQuant software platform[52] version 1.6.1. 0 was used to process raw files and search was conducted against the *Saccharomyces cerevisiae* complete/reference proteome database (Uniprot, released in September 2017), using the Andromeda search engine[53]. iBAQ (intensity-based absolute quantification) values calculated by MaxQuant are the (raw) intensities divided by the number of theoretical peptides (Fig. 1c and Supplementary Data 2). In that way, iBAQ values are proportional to the molar quantities of the proteins[28].

Relative molar abundances, for each protein, was determined as its relative iBAQ (riBAQ), a normalized measure of molar abundance. We divided each yeast protein's iBAQ value by the sum of all non-contaminant iBAQ values[54]. Enrichment was calculated as the riBaq ratio between protein pull-down and negative control (Fig. 1c and Supplementary Data 3). All proteins showing a ratio inferior at 2 was considered as not enriched.

**CRAC**. CRAC was performed as described[55] on yeast strains expressing the protein of interest carrying a C-terminal HTP tag, grown in SD medium to log phase and UV crosslinked (254 nm, 100 s) to covalently bind RNA to protein. RNA–protein complexes have been purified, and RNAs are partially digested to leave only the "footprint" of the protein or protein complex. Mircat and barcoded linkers (containing three random nucleotides) have been ligated on both 3′ and 5′ end, respectively. Proteins were then digested by proteinase K, RNAs were reverse transcribed, and PCR amplified. cDNA libraries were size fractionated on agarose gels and then subjected to next-generation sequencing using Illumina HiSeq (Edinburgh Genomics) or Illumina Miniseq (our laboratory). Illumina sequence data from this publication have been submitted to the GEO database (http://www.ncbi.nlm.nih.gov/geo/) and assigned the identifier GSE135526.

**RNAseq**. Yeasts were grown at 30 °C to OD 0.5 in minimal media and RNAs were extracted using a standard acidic hot phenol method. RNAseq libraries were prepared using a SENSE mRNA-seq Library Prep Kit V2 for Illumina (Lexogen), as recommended by the manufacturer. Two micrograms of RNA was denatured and used as input for poly(A)-tailed RNA purification through hybridization on oligodT magnetic beads. Purified poly(A) RNA were subjected to reverse transcription and ligation generating short cDNA fragments with linker sequences at either end. The library was converted to double-stranded DNA, purified, and PCR amplified (13 cycles). Samples were checked on bioanalyser and then sequenced using standard Illumina protocol on NextSeq (75 cycles, high output).

**Real-time PCR amplification**. Yeasts were grown at 30 °C to OD 0.5 in minimal media and RNAs were extracted using a standard acidic hot phenol method. Five micrograms of 5 μg RNA was submitted to TURBO DNase treatment (Thermo), checked for quality, and reverse transcribed with Random decamers (Ambion) and Superscript III Reverse Transcriptase (Invitrogen) following the manufacturer's protocols. After RNAse H (Thermo Scientific) treatment, cDNAs were diluted 20-fold. Real-time PCR amplifications were performed in a LightCycler480 system (Roche Diagnostics) in 384-well plates on 5 μL scale reactions using 2 μL diluted cDNAs and Brilliant III Ultra-fast SYBR Green qPCR Master Mix (Agilent) with primers listed in Supplementary Data 7 at a final concentration of 0.4 μM. Technical triplicates for each independent biological replicates (two for WT and three for *trf5Δ*) were carried out. For each sample, to ensure the level of residual genomic DNA (gDNA) in the RNA prep were negligible, qPCR was performed without reverse transcription on DNase-treated RNA ("no RT"). Primer's linearity and efficiencies were tested by performing a standard amplification on serial dilution of WT gDNA in triplicates (100, 10, 1, 0.1, and 0.01 ng μL⁻¹). Cycle threshold (Ct)

values were averaged between triplicates for each RNA sample. Gene expression fold change were determined as reported[56]. For qPCR raw data, primer sequences and efficiencies, and fold change calculation on WT and trf5Δ strains, see Supplementary Data 7.

**CRAC data: pre-processing and alignment**. CRAC sequencing data were quality filtered and adapter were trimmed using Flexbar 3.4.0[57] with parameters -x 1 –ao 4 -g and only reads containing the 3′ adapter were kept. Then, the sequences were collapsed: reads having identical ends and identical random nucleotides in the 5′ barcode were counted as one, allowing removal of PCR duplicates. CRAC pre-processed reads containing non-encoded oligo-A tailed were identified using a pipeline developed by Grzegorz Kudla[10,41]. Reads were then aligned to the *Saccharomyces cerevisiae* genome (SGD v64) using Novoalign (Novocraft, V2.07) with genome annotation from Ensembl (EF4.74)[58], supplemented with non-coding sequences as described in ref. [41], with parameters –r Random.

**RNAseq data: pre-processing and alignment**. Low-quality reads and 5′ extremities of reads were filtered out. Low-complexity sequences (reads having more than 70% of their content corresponding to a single-nucleotide stretch and that would be potentially misaligned) were filtered out before alignment. Then, RNAseq reads were aligned with STAR[59] using genome database from Ensembl (EF4.74).

**Class distribution**. Downstream analyses were performed using pyCRAC software (0.5.3)[60]. pyReadCounters (pyCRAC) was used to count overlaps with genes and reads per million (RPMs) or RPM per kilobase (RPKM).

**Correlation of binding**. For correlation of binding analyses, low-complexity sequences were filtered out before alignment. Overlaps either with transcripts (all or RNAPII transcripts when stated) or with a genome reference file divided into 50 nt windows, independently of transcripts, were calculated, averaged between two biological repeats, and used to calculate Pearson's correlations between samples.

**Plots, binding profiles**. Plot showing binding along single genes were generated using pyPileup (pyCRAC). Metagenomic plots were performed using homemade script using pyPileup on each individual transcript to count hits at each position. We obtain a table in which each row represents a transcript and each column represents the absolute position from 5′ end or 3′ end. The plot is summing up binding at each position allowing the display of a binding profile aligned either at 5′ end or 3′ end (Fig. 4). To calculate the enrichment or loss of mRNA signal between WT strain and trf5Δ strain, normalized coverage (RPM) was calculated at each position along the genome for all datasets and averaged between replicate RNAseq datasets (four and three independent replicates for WT and trf5Δ, respectively). Log 2 enrichment was calculated after the addition of five pseudocounts to both the numerator and denominator as described[61].

**Clustering analysis of mRNAs**. We selected and combined the top 1000 protein-coding genes bound by each TRAMP component (2005 mRNAs were included). pyBinCollector (pyCRAC) was used to calculate binding distribution along transcripts, each one being divided into five bins from TSS to pA site. Binding across each bin was calculated as a fraction of total binding across individual gene (set to 1). For each transcript, we averaged the number of reads between two biological replicates to reduce the influence of experimental variation upon clustering analysis. We normalized the data for each gene. Air1, Air2, Trf4, and Mtr4 coverage along transcripts were then clustered using cluster 3.0 (C Clustering Library version 1.52a, k-medians, k = 6, Euclidean distance) and data displayed as heat maps. For data not included in the clustering analysis, pyBincollector output was sorted according to the clustered list and shown as heat map.

**Peak calling**. Low-complexity sequences trimmed dataset were used. The number of hits for each position of the genome was calculated excluding all non-RNA polymerase II transcripts and normalized to RPMs of RNA polymerase II transcripts. Clusters consisting of 15 continuous position with more than 50 hits across were detected. The highest position in each cluster was selected as "peak." Air1 peaks were used as a reference file to which other protein peaks were aligned (Fig. 6b–m and Supplementary Fig. 10).

**Reporting summary**. Further information on research design is available in the Nature Research Reporting Summary linked to this article.

## Data availability

All sequence data from experiments made for this study and Rat1 datasets from Granneman et al.[62] are available from GEO under accession number GSE135526. The mass spectrometry proteomics data have been deposited to the ProteomeXchange Consortium via the PRIDE partner repository with the dataset identifier PXD017114. Published data are available under GEO accession number GSE77683 (Mtr4), GSE69696 (Rrp44, Rrp6, Air2), GSE79950 (UTP proteins), GSE70191 (Nab3, Npl3, Sto1),

GSE114680 (Nop1, Nop56, Nop58), and GSE46742 (Trf4, Cbc1, Gbp2, Tho2, Hrp1, Nab2, Pab1, Mex67, Hek2, Xrn1, Ski2). We used *Saccharomyces cerevisiae* genome version (SGD v64) with genome annotation from Ensembl (EF4.74) for analysis. The source data underlying Figures and Supplementary Figures are provided as a Source Data file. All data are available from the corresponding author upon reasonable request. Source data are provided with this paper.

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

## Acknowledgements

We thank Stefan Bresson, Marie Winz, and Aleksandra Helwak for critical reading of the manuscript. The Wellcome Trust generously funded this work through a Principle Research Fellowship to D.T. (077248), a Senior Research Fellowship to J.R. (103139), and an instrument grant (108504). C.D.-F. was supported by an FEBS Long-Term Fellowship. Work in the Wellcome Center for Cell Biology is supported by a Center Core grant (203149).

## Author contributions

C.D.-F. and D.T. conceived the experiments, analyzed the data. and wrote the manuscript. C.D.-F. and C.S. performed the experiments. C.S. and J.R. participated in proteomic analyses. All authors edited and reviewed the manuscript.

## Competing interests

The authors declare no competing interests.
