## [Peer Review File · Nature Communications]

Reviewers' comments:

Reviewer #1 (Remarks to the Author):

General comments:

In this very interesting manuscript by Delan-Forino et al., the authors gain insights into the TRAMP complex and its substrate recognition in vivo. The manuscript contains many experiments and a wealth of data and thus I found the manuscript not always easy to read, as it is sometimes very methodical/technical.

S. cerevisiae has two paralogs of the non-canonical poly(A) polymerases Trf4/5 and two paralogs of the Zn-knuckle proteins Air1/2. Thus there are there are four potential combinations possible TRAMP4-1, 4-2, 5-1 and 5-2. They authors set first out to see which TRAMP combinations do occur in vivo and in the following they try to delineate what the substrate specificity of the different TRAMP complexes are.

They use a number of methods including protein affinity purification together with mass spectrometry, RNA cross linking, RNA-sequencing and bioinformatics to understand how TRAMP is targeted to substrates and if the different TRAMP complexes have different substrates specificity.

For example, the authors show that Trf5-Air1 is involved in ITS1 pre-rRNA processing pathway, whereas Trf4-Air2 seems to be enriched/ associated with RNAPII transcripts close to the TSS.

Overall the data are of good quality and contain the necessary controls. However, for several points I think the data do not fully support the conclusions of the authors. For example, initially in Figure 2A they show that Air1 has a clear preference for ETS/ITS regions of rRNA, however, in the discussion they conclude that the substrate selection is mainly performed by the Trf4/5 proteins. I could not follow how they came from one to the other conclusion.

Major points:

1. Air1/2 and Trf4/5 are paralogs of each other and the authors state in the introduction that the core regions of both protein are more similar and the N-terminal parts are more divergent. Thus, I would have expected that the authors perform some domain swapping experiments, for example take the N- and C-terminus of Trf5 add them to Trf4 and vice versa take the appendices from Trf4 and put them on Trf5 and then compare this in the CRAC experiments. The same could be done for Air1/2. This could give interesting insights into the targeting mechanism of TRAMP. Would this be possible to do?

2. It is well known that Trf4/5 and Air1/2 directly bind each other. But interestingly, the proteins with highest abundance in the Air1-pulldown are ribosomal proteins and not factors of TRAMP (Table S2). The situation is similar in Trf5-pulldown. The bait protein is found at position 61 and prey protein Air1 at position 211 when proteins are sorted by abundance. Here, too, the factors with the highest abundance are ribosomal factors. I am not an expert in mass spectrometry but it would be helpful if the authors could comment on this approach for the determination of in vivo occurring TRAMP combinations. Perhaps this should also be mentioned in the text or in the methods. It is however very interesting, that Air2 seems to exclusively bind Trf4 and Trf5 in turn exclusively binds Air1. Do the authors have an idea how this selectivity might be achieved? Maybe it is also worth to consider the absolute copy number of TRAMP components that showed that Trf4 is 4-times more abundant than Trf5 and Air2 2-fold more than Air1. (Kulak et al. Nat Methods 2014). And also here the domain swapping experiments could

reveal an interesting mechanism of how the individual pairs form.

3. In Table S2 a few gene names are missing in column 1, for example in lines 178, 218, 305 and 455. Furthermore I can only find Air1, Trf4 (PAP2) and Trf5 in the list of proteins. Air2 (YDL175C) is not in the list and so I cannot calculate the values shown in Figure 1C.

4. In the CRAC Analysis the authors assign RNA substrates bound to the individual TRAMP components. A) For rRNA they analyse for the presence of ETS/ITS, indication that these are pre-rRNAs. As TRAMP is nuclear, one would also expect that that TRAMP binds pre-mRNAs during the surveillance process. Did the authors also check for the presence of introns in the mRNA fraction and if yes, what is the conclusion?

5. ITS1 is a substrate of Trf5/Air1:

a. In Figure 3 it would help the reader to compare the data if the scale would be the same for all panels in Figure 3. In panels A-D the maximum is 10k, in E 50 k, in F-J 20k, and in K 10k. I would suggest that the authors quantify the height of the main peak and show it as a bar diagram that allows direct comparison without taking a ruler.

b. When checking for protein binding to ITS1 the authors find Air1 and Trf5 are present but no binding of Air2 and Trf4. Figure 3E shows that for Mtr4 on average there is a signal of ~38k HPM and upon deletion of Trf5 is reduced to ~7k HPM (3I) and upon deletion of Air1 to ~10k HPM (3F). However, deletion even the deletion of Trf4, which does not bind ITS1, reduces the signal for Mtr4 by a factor of two (3H) and Air2 by a factor of four (3G), similar to the deletion of Air1. The authors then conclude that deletion of Trf5 strongly reduces Mtr4 binding (my calculation: 5-6-fold), whereas Air1 had a more modest effect (my calculation: 4-fold). The interpretation then is that Air1 could be replaced by Air2 and thus the effect is less pronounced. However, this seems, to my understanding, to be in contrast to the data shown in Figure 2A, which shows that Air1 seems to be the main factor associated with pre-rRNA (ETS/ITS) regions. This interpretation would also not agree with Figure 3B and 3C that shows that Trf4 and Air2 do not bind ITS1. An alternative explanation could be that the binding/recruitment of Mtr4 to ITS1 is too large extents not dependent on the Air and Trf proteins, as already the deletion of the Mtr4 arch strongly reduces Mtr4 binding and the arch is not required to form TRAMP. But the overall stability/residency of Mtr4 could be higher within TRAMP as multiple components (Mtr4, Air1, Trf5) forms contacts with the ITS1 RNA and thereby achieve cooperativity.

c. Therefore the 'mechanistic' scheme in Figure 3M implies that Trf5/Air1 bind first and then recruit Mtr4, however, this is in my opinion not supported by the data. To distinguish the different scenarios the authors could utilise strains in which all components are present but the formation of TRAMP is impaired. As the binding site of Trf4/Air2 on Mtr4 are known, either a Mtr4 mutant that can not form TRAMP or instead a strain in which the Mtr4 binding sites within Trf5/Air1 have been mutated could be used for this.

d. Another question that I have is: What is the cutoff between strongly reduced and modest in fold-changes that the authors apply? Given that the level of Trf5 binding seems to be quite variable (panel 3D), I would suggest the authors quantify the data for Mtr4 in different mutant background with fold-changes. This will allow the reader to compare the changes much better on a quantitative level.

6. In the section 'Loss of Trf5 alters Mtr4 association and abundance of targeted mRNAs' write: However, the heat maps (Fig. 5) indicated that subsets of nuclear mRNAs show preferential enrichment for binding to Trf5. It is not clear to me to which cluster/subset are the authors referring in Figure 5A. Do these nuclear mRNAs form/belong to any cluster and could this be mentioned in the text? On page 11 the authors write: A notable finding from the heat maps was the identification of group of mRNAs showing strong 3' association specifically with Trf5 (cluster 1 in Fig. 5). Other TRAMP and exosome components were not colocalized on these mRNAs, suggesting a distinct function for Trf5. By looking at Figure 5A on Cluster 1, I agree that Trf5 is clearly enriched on the 3' end of the mRNA close to the pA site. Two things are surprising to me: A) If Trf5 forms a complex with Air1, why is Air1 then not enriched on those transcripts? B) At the same time Air2, but to lesser extent also Mtr4, are enriched on cluster 1, however, near the TSS at the 5' end. Mtr4 binding to these messages was neither affected by deletion of Trf4 nor Trf5, but Air2 is clearly present on these mRNAs. How do the authors interpret this scenario with Air2 being enriched at near the TSS and Trf5 near the pA site.

Minor comments:

Figure 1A: This panel is too crowded and the reader does not know where to focus on. I would suggest to simplify the color scheme of the alignment itself by reducing it to one colour (e.g. blue) and use shades to show different degrees of conservation. It is in my opinion not necessary to color the different amino acid by their physico-chemical properties.

In Figure 3 A-K: Do the dark and light colored lines represent experimental replicates? This is not explained in the figure legend.

Figure 6 is mislabelled. Panel E occurs twice and in for Nab2 in the column it should be panel G. Similarly, on page 10, last paragraph needs to be corrected: which includes the abundant snoRNA species (Fig. 6B – 6E), or considering only mRNAs (Fig. 6F – 6I). I assume this should reference to (Fig. 6B-G) and then (Fig. 6H-M). Then the authors mention "Close colocalization was seen for Air1 and Air2 (Fig. 6E and 6K), supporting the conclusion that they are functionally redundant on many substrates. However, Air1 is not shown in the figure at all. It only displays TRAMP components Trf4, Trf5 and Air2. Could the authors please clarify this.

Page 12, last sentence: When bound to the exosome, Mtr4 is positioned at the entrance to the central lumen, such that RNAs can pass through the helicase domain directly into the exosome. Please cite the literature that demonstrates the Mtr4 – exosome interaction. For yeast Mtr4 it was first shown by the group of Elena Conti (Schuller et al.) and for human Mtr4 by Chris Lima's lab (Weick et al.).

Reviewer #2 (Remarks to the Author):

The manuscript entitled "Substrate Specificity of the TRAMP Nuclear Surveillance Complexes" by Delan-Forino et al. provides a novel and extensive analysis the protein-protein and protein-RNA interactions of Trf4/5-Air1/2-Mtr4 polyadenylation (TRAMP) complexes - key cofactors for the RNA exosome - in *S. cerevisiae*. In particular, proteomic analysis of Air1, Air2, Trf4, and Trf5 reveals three distinct TRAMP complexes in vivo - TRAMP4-1 (Trf4, Air2), TRAMP4-2 (Trf4, Air1), and TRAMP5 (Trf5, Air1) - that each

likely associate with Mtr4 and make differential interactions with surveillance factors, pre-mRNA processing factors, snoRNPs, and ribosome biogenesis factors. In addition, CRAC analysis (UV-crosslinking and sequence analysis of cDNA) of Air1, Air2, Trf4, Trf5, Mtr4, and Mtr4 Δ arch, an Mtr4 mutant lacking the arch domain implicated in substrate recruitment, reveals that the TRAMP complexes have distinct substrate preferences. Notably, Air1 and Trf5 show significant binding to pre-rRNA spacer regions (ETS, ITS), whereas Air2 and Trf4 show significant binding to ncRNAs (CUTs, SUTs, XUTs). Moreover, Air2 and Trf4 show enrichment at 5' promoter-proximal regions of pre-mRNAs, whereas Air1 and Trf5 show enrichment at 3' pA site-proximal regions of pre-mRNAs. Surprisingly, on many RNA substrates, binding specificity is conferred by Trf4 and Trf5 poly(A) polymerases, as opposed to the Air1 and Air2 zinc knuckle RNA binding proteins. Trf5 binding to mRNAs is also found to be important for recruitment of Mtr4 and control of mRNA levels. Overall, this work provides novel insights into the function of the TRAMP complexes and sheds some much-needed light on the substrate specificity of these critical exosome cofactors. However, at present, the manuscript requires a few additional experiments to support the results.

Major Points:

1. To support Figure 1E, the authors should validate a few of the Trf4/5 and Air1/2 interacting proteins identified by tandem-affinity purification and mass-spectrometry using purification and Western blotting. In particular, validation of the binding of Trf4, Air1, and Air2 to the surveillance factor, Xrn1, and one snoRNP protein, e.g. Snu13 or Gar1, would be important.
2. In Figure 3, the high association of Trf5, Air1, and Mtr4, but not Trf4 and Air2, within the A2-A3 region of ITS1 pre-rRNA and loss of Mtr4 association upon deletion of TRF5 suggests that degradation of 23S pre-rRNA/ITS1 pre-rRNA should be more affected in *trf5 Δ and *air1 Δ cells compared to *trf4 Δ and *air2 Δ cells. The authors should test this possibility by examining the levels of 23S rRNA/ITS1 pre-rRNA in *trf5 Δ , *air1 Δ , *trf4 Δ , and *air2 Δ cells by Northern blotting. If a 23S pre-rRNA processing defect is observed in *trf5 Δ and *air1 Δ cells, the authors should test whether overexpression of Mtr4 rescues the defect.**********
3. To support Figure 7, the authors should validate some the mRNA targets that increase in *trf5 Δ cells by RT-qPCR. In addition, given that Mtr4 association with Trf5 mRNA targets is reduced in *trf5 Δ cells, suggesting decreased recruitment of Mtr4 upon loss of Trf5 leads to increased levels of mRNAs, the authors should test whether overexpression of Mtr4 can reduce the levels of the mRNAs in *trf5 Δ cells. Although not essential, it would interesting to know whether the increase in mRNA levels observed in *trf5 Δ cells can be reduced by a catalytically inactive mutant of Trf5.****

Minor Points:

1. p.3-128 - "Mtr4" should be "MTR4" to denote human protein.
2. p.3-30 - "NEXT and PAXT" acronyms should be defined.
3. p.5-1.30 - "Nrd1-intereacting" should be "Nrd1-interacting."

4. p.7-l.24 - "23S RNA" should be "23S rRNA."
5. p.9-l.30 - Should there be a reference to Fig. 4R model?
6. p.9-l.24 - Insert comma. "Trf4, Trf5 Air1..." should be "Trf4, Trf5, Air1..."
7. p.13-l.5 - "conceivably extent..." should be "conceivably extend..."
8. p.13-l.6 - define "symmetry-related molecule."
9. p.13-l.29 - replace "effects of the trf5 Δ on" with "effects of deletion of TRF5 on."
10. Figure 1C - define units of y-axis.

Reviewer #3 (Remarks to the Author):

In this manuscript Delan-Fiorino and colleagues report the distribution of TRAMP and exosome components on different classes of RNA in vivo using a technique called CRAC (Crosslinking analysis of cDNAs), similar to iCLIP. The authors first define the composition of alternative TRAMP complexes by quantitative mass spectrometry, using mutants and different purification conditions. In the second part of the manuscript, the authors analyze the RNA occupancy of these factors in wild type and mutant conditions. Based on these analyses, they propose a model to explain the specificity of TRAMP binding to its different substrates. They also report the unexpected binding of Trf5 to the 3' end of a set of mRNAs.

In spite of the large amount of work that it represents, this study falls short of providing the significant conceptual and mechanistic advance that would be required for publication in Nature Comm. The work remains highly descriptive, with some conclusions that are suggested but not proven by the data. The finding that Trf5 occupancy peaks at the 3' end of some mRNAs is novel and potentially very interesting, but it is poorly exploited in terms of mechanistic analysis.

Specific points:

- For all CRAC analysis using mutants (e.g. Trf4 Δ arch or Mtr4 in strains deleted for other TRAMP components), it should be shown that the abundance of the protein of interest is not altered by the mutation or the mutant background. Ideally in the same CRAC experiment
- In figure 3 the authors show the effect of deleting TRAMP components on the recruitment of Mtr4 to the ITS1. The authors should assess the statistical significance of the differences they see, considering the differences seen in some replicates (e.g. Trf5 and Mtr4). For instance, it appears from panel H that deletion of Trf4 leads to a reduction of roughly 50% of the Mtr4 signal (depending on the Mtr4 replicate). Is this significant? The authors conclude that there is only a minor effect of deleting Trf4, but it is unclear on which basis. Same for Air1/2: deletion of either protein has the same negative effect on Mtr4 binding, still only Air1 is considered to have a significant effect: why is so? Air2 and Trf4 do not bind this region at all (panels B-C), so it is unclear how they could be involved in Mtr4 recruitment. Still, upon their deletion, Mtr4 recruitment is affected, in the case of Air2 to similar levels as for deletion of Air1. The conclusion that Trf5-Air1 recruits Mtr4 to ITS1 (title of the section) is not justified in the light of

the effects seen with Air2 and Trf4. Moreover, a decrease in RNA occupancy is not formal proof of a recruitment defect.

- FigS4: The significance of this figure is unclear. The authors compare the effect of deleting the arch domain to the effect of deleting Trf5 on the recruitment of Mtr4 (panel A). In a similar plot (C) they also look at the effect of $\text{trf4}\Delta$ versus $\text{trf5}\Delta$. The scatter plots do not show clear correlations between the datasets. The authors conclude that Δarch has a stronger effect than $\text{trf5}\Delta$ and $\text{trf4}\Delta$ has a stronger effect than $\text{trf5}\Delta$, but there is only a qualitative, visual assessment of these differences. There might be more genes responsive to Δarch than Δtrf5 (S4A) but this is not evaluated statistically. Also by visual assessment there are as many, or possibly more genes for which the binding of Mtr4 is actually increased in a Δarch (panel A, left of the y-axis) and certainly more genes with increased Mtr4 binding in the absence of Trf5 (panel C, below the x-axis): why these are ignored ?

- The metagene analyses shown in figure 4 should be repeated using the median instead of the mean, to avoid effects that might be due to outliers. This, for instance, occurs for the 3'-end peak of Air1 that only is detected at a minority of genes (maybe 10%, group 2). Also, the strong difference between the replicates is a major concern here.

- The 3' end occupancy peak observed for Trf5 is very convincing (although here too the replicates are very different). This potentially important finding is not really investigated further, except for one RNAseq analysis that detects a decrease in 3'-end reads for the genes of group 2: but what are these 3'-end truncated transcripts ? Different isoforms because of alternative polyadenylation ? Products of partial degradation ? These are important questions that, regrettably, are not addressed.

- The conclusion that substrate specificity is provided by Trf4 and Trf5 is only based on occupancy effects in mutants, which can be very indirect. At least some in vitro RNA binding experiments should be provided to support these conclusions: are Trf4 and Trf5 able to bind RNA specifically and independently of Air1/2 ?.

Minor points:

- Detection of Xrn1 association with TRAMP, even though RNase sensitive, is surprising. The authors should consider the possibility that Xrn1 associates in the extract, after cell breakage.

We thank the referees for their careful attention in reviewing the MS. In response to their helpful comments we have added further analyses and experiments, which are presented in three additional supplementary figures. While these have not changed our major conclusions, we feel that they have strengthened key points in the work and clarified these for the reader.

Detailed responses are given below.

Reviewer #1 (Remarks to the Author):

General comments:

In this very interesting manuscript by Delan-Forino et al., the authors gain insights into the TRAMP complex and its substrate recognition in vivo. The manuscript contains many experiments and a wealth of data and thus I found the manuscript not always easy to read, as it is sometimes very methodical/technical

We have attempted to improve the readability of the revised MS and have made text changes in many places.

S. cerevisiae has two paralogs of the non-canonical poly(A)polymerases Trf4/5 and two paralogs of the Zn-knuckle proteins Air1/2. Thus there are there are four potential combinations possible TRAMP4-1, 4-2, 5-1 and 5-2. They authors set first out to see which TRAMP combinations do occur in vivo and in the following they try to delineate what the substrate specificity of the different TRAMP complexes are. They use a number of methods including protein affinity purification together with mass spectrometry, RNA cross linking, RNA-sequencing and bioinformatics to understand how TRAMP is targeted to substrates and if the different TRAMP complexes have different substrates specificity.

For example, the authors show that Trf5-Air1 is involved in ITS1 pre-rRNA processing pathway, whereas Trf4-Air2 seems to be enriched/ associated with RNAPII transcripts close to the TSS.

Overall the data are of good quality and contain the necessary controls. However, for several points I think the data do not fully support the conclusions of the authors. For example, initially in Figure 2A they show that Air1 has a clear preference for ETS/ITS regions of rRNA, however, in the discussion they conclude that the substrate selection is mainly performed by the Trf4/5 proteins. I could not follow how they came from one to the other conclusion.

We predict that the differences in Air1 distribution largely reflects its affinity for Trf5. We have altered the text to make this clearer (p8).

Major points:

1. Air1/2 and Trf4/5 are paralogs of each other and the authors state in the introduction that the core regions of both protein are more similar and the N-terminal parts are more divergent. Thus, I would have expected that the authors perform some domain swapping experiments, for example take the N- and C-terminus of Trf5 add them to Trf4 and vice versa take the appendices from Trf4 and put them on Trf5 and then compare this in the CRAC experiments. The same could be done for Air1/2. This could give interesting insights into the targeting

mechanism of TRAMP. Would this be possible to do?

We indeed attempted to perform these experiments for Trf4 and Trf5. Unfortunately, the N-terminal domain swaps did not result in stable protein accumulation. Swapping the C-terminal domains did generate stable fusion proteins, which were analyzed by CRAC. These did not give very strong crosslinking, resulting in poor signal to noise ratios and consequent variability in results, but we have included the data obtained (new Figures S5 and S8).

2. It is well known that Trf4/5 and Air1/2 directly bind each other. But interestingly, the proteins with highest abundance in the Air1-pulldown are ribosomal proteins and not factors of TRAMP (Table S2). The situation is similar in Trf5-pulldown. The bait protein is found at position 61 and prey protein Air1 at position 211 when proteins are sorted by abundance. Here, too, the factors with the highest abundance are ribosomal factors. I am not an expert in mass spectrometry but it would be helpful if the authors could comment on this approach for the determination of in vivo occurring TRAMP combinations. Perhaps this should also be mentioned in the text or in the methods. It is however very interesting, that Air2 seems to exclusively bind Trf4 and Trf5 in turn exclusively binds Air1. Do the authors have an idea how this selectivity might be achieved?

Ribosomal subunits are major contaminants in almost all analyses involving precipitation and MS. This is not easily resolved, and ribosomal proteins have simply been omitted from many published analyses - both proteome-wide and specifically for pre-ribosomes, perhaps giving a misleading impression.

As an example: In Thoms et al. (2015), the authors carried out similar mass spectrometry analysis on Mtr4 pull-downs and similar high abundance contaminants were recovered (ribosomal proteins and chaperones). TRAMP components were ranked lower than some contaminants: in data ranked by iBaq, Trf4 was position 12, Air2 was 20th, Air1 was 53th and Trf5 65th. Other known Mtr4 interactors are in position 80 for Nop53 and position 114 for Utp18.

In our pull downs, several top ranked proteins are highly abundant and were also recovered in the no bait negative control (column L and M), thus being probable contaminants.

Maybe it is also worth to consider the absolute copy number of TRAMP components that showed that Trf4 is 4-times more abundant than Trf5 and Air2 2-fold more than Air1. (Kulak et al. Nat Methods 2014).

The referee raises an interesting point. Slightly different relative abundances are given by SGD, based on the median values from 14 different publications:

Air1: Median 1851 +/- 466

Air2: Median 1750 +/- 420

Trf4: Median 2659 +/- 411

Trf5: Median 1329 +/- 608

We mention these values in the revised MS (p13). These are entirely consistent with our conclusion that Trf4 interacts with Air1 and Air2, whereas Trf5 binds only Air1.

This might suggest the very approximate abundances:

Trf5-Air1; 1,300 copies per cell

Trf4-Air2; 1,700 copies per cell

Trf4-Air1; 600 copies per cell

And also here the domain swapping experiments could reveal an interesting mechanism of how the individual pairs form.

As noted above, these experiments were attempted, and the results obtained are included in the revised text.

3. In Table S2 a few gene names are missing in column 1, for example in lines 178, 218, 305 and 455. Furthermore I can only find Air1, Trf4 (PAP2) and Trf5 in the list of proteins. Air2 (YDL175C) is not in the list and so I cannot calculate the values shown in Figure 1C.

This has been corrected in the revised Table S2.

4. In the CRAC Analysis the authors assign RNA substrates bound to the individual TRAMP components. A) For rRNA they analyse for the presence of ETS/ITS, indication that these are pre-rRNAs. As TRAMP is nuclear, one would also expect that that TRAMP binds pre-mRNAs during the surveillance process. Did the authors also check for the presence of introns in the mRNA fraction and if yes, what is the conclusion?

As the referee predicts, all TRAMP components showed substantial numbers of introns hits. TRAMP4-1 appears to be the TRAMP complex that most strongly targets introns. Notably, the Arch domain does not appear to be required for Mtr4 recruitment on introns. This result is shown in new Figure S4 and described on p9.

5. ITS1 is a substrate of Trf5/Air1:

a. In Figure 3 it would help the reader to compare the data if the scale would be the same for all panels in Figure 3. In panels A-D the maximum is 10k, in E 50 k, in F-J 20k, and in K 10k. I would suggest that the authors quantify the height of the main peak and show it as a bar diagram that allows direct comparison without taking a ruler.

New panels (Figure 3M and 3N) have been added in Figure 3 in which total reads encompassing the region from A2 to A3 cleavage sites (+ 10nt on each side) were summed for each individual replicate. An average was calculated for biological replicates and the standard deviation is shown to facilitate direct comparison between different TRAMP components in wild-type and mutant backgrounds.

b. When checking for protein binding to ITS1 the authors find Air1 and Trf5 are present but no binding of Air2 and Trf4. Figure 3E shows that for Mtr4 on average there is a signal of ~38k HPM and upon deletion of Trf5 is reduced to ~7k HPM (3I) and upon deletion of Air1 to ~10k HPM (3F). However, deletion even the deletion of Trf4, which does not bind ITS1, reduces the signal for Mtr4 by a factor of two (3H) and Air2 by a factor of four (3G), similar to the deletion of Air1. The authors then conclude that deletion of Trf5 strongly reduces Mtr4 binding (my calculation: 5-6-fold), whereas Air1 had a more modest effect (my calculation: 4-fold). The interpretation then is that Air1 could be replaced by Air2 and thus the effect is less pronounced. However, this seems, to my understanding, to be in contrast to the data shown in Figure 2A, which shows that Air1 seems to be the main factor associated with pre-rRNA (ETS/ITS) regions.

This interpretation would also not agree with Figure 3B and 3C that shows that Trf4 and Air2 do not bind ITS1. An alternative explanation could be that the binding/recruitment of Mtr4 to ITS1 is too large extents not dependent on the Air and Trf proteins, as already the deletion of the Mtr4 arch strongly reduces Mtr4 binding and the arch is not required to form TRAMP. But the overall stability/residency of Mtr4 could be higher within TRAMP as multiple components (Mtr4, Air1, Trf5) forms contacts with the ITS1 RNA and thereby achieve cooperativity.

The referee is correct in noting that loss of Air2 results in defects, some of which are almost as strong as for Air1, and we discuss this in the revised manuscript. Our results strongly suggest that TRAMP complex stability and balance is modified when one component is deleted. We observed in Figure 1 that Trf5 binds to Air2 when Air1 is deleted, whereas almost no binding was seen in the in wild-type. We suggest that in absence of Air2, Air1 is more frequently bound by Trf4 than in the WT, reducing its availability to form TRAMP5-1 and target ITS1. Deletion of Trf4 has a much milder effect than deletion of Trf5 on Mtr4 binding across A2-A3 region (2-fold for *trf4* Δ compared to 5.8-fold for *trf5* Δ). This also can be explained by the redundancy of the TRAMP components; in the absence of Trf4, some Trf4-binding sites will be bound by Trf5, resulting in a loss in targeting of Trf5-specific substrates. Were Air1 more important than Trf5 for recruitment, only a mild decrease of Mtr4 binding in *trf5* Δ (and *air2* Δ) should be observed since TRAMP4-1 should then replace it; however, this is not the case.

c. Therefore the 'mechanistic' scheme in Figure 3M implies that Trf5/Air1 bind first and then recruit Mtr4, however, this is in my opinion not supported by the data. To distinguish the different scenarios the authors could utilise strains in which all components are present but the formation of TRAMP is impaired. As the binding site of Trf4/Air2 on Mtr4 are known, either a Mtr4 mutant that can not form TRAMP or instead a strain in which the Mtr4 binding sites within Trf5/Air1 have been mutated could be used for this.

The scheme has been redrawn in the revised figure and is now shown as Fig. 3P and clearly labeled as a "Model" in the legend. It is known that the association of Trf5-Air1 with this region does not require Mtr4, since depletion of Mtr4 leads to hyper-adenylation of the A3-cleaved 23S RNA, presumably because handover from TRAMP5-1 to the exosome is blocked¹. The experiments proposed by the referee are certainly worthwhile in understanding the details of the TRAMP => exosome handover, for this and many other substrates, but we feel that they are beyond the scope of the current MS.

d. Another question that I have is: What is the cutoff between strongly reduced and modest in fold-changes that the authors apply? Given that the level of Trf5 binding seems to be quite variable (panel 3D), I would suggest the authors quantify the data for Mtr4 in different mutant background with fold-changes. This will allow the reader to compare the changes much better on a quantitative level.

A new panel has been added to Figure 3, in which total reads encompassing the region between cleavage sites A2 and A3 plus 10 flanking nts, were summed for each individual replicate. The manuscript text has been modified consequently. The average was calculated for biological replicates and standard deviations are shown to facilitate direct comparison between different TRAMP components in wild-type and mutant backgrounds. Mtr4 binding to the A2-A3 region is reduced by:

4.4-fold for *mtr4* Δ *arch*

3.7-fold for *air1*Δ
3.2-fold for *air2*Δ
2.0-fold for *trf4*Δ
5.8-fold for *trf5*Δ.

6. In the section 'Loss of Trf5 alters Mtr4 association and abundance of targeted mRNAs' write: However, the heat maps (Fig. 5) indicated that subsets of nuclear mRNAs show preferential enrichment for binding to Trf5. It is not clear to me to which cluster/subset are the authors referring in Figure 5A. Do these nuclear mRNAs form/belong to any cluster and could this be mentioned in the text?

This referred to clusters 1 and 2 and we have clarified the revised text (p11).

On page 11 the authors write: A notable finding from the heat maps was the identification of group of mRNAs showing strong 3' association specifically with Trf5 (cluster 1 in Fig. 5). Other TRAMP and exosome components were not colocalized on these mRNAs, suggesting a distinct function for Trf5. By looking at Figure 5A on Cluster 1, I agree that Trf5 is clearly enriched on the 3' end of the mRNA close to the pA site.

Two things are surprising to me: A) If Trf5 forms an complex with Air1, why is Air1 then not enriched on those transcripts?

Trf5 is stably bound to Air1 in cell lysates, but we do not know that all Trf5 is present in this complex *in vivo*. Unfortunately, apart from the data presented here, there is no simple way to determine the proteins associated with Trf5 when it is bound at any specific site or set of sites on endogenous RNAs. The comparisons of protein binding sites shown in Figures 6 and S2 are first attempts to address this issue.

B) At the same time Air2, but to lesser extent also Mtr4, are enriched on cluster 1, however, near the TSS at the 5' end. Mtr4 binding to this messages was neither affected by deletion of Trf4 nor Trf5, but Air2 is clearly present on these mRNAs. How do the authors interpret this scenario with Air2 being enriched at near the TSS and Trf5 near the pA site.

In cluster 1, mRNAs are very likely targeted by 2 distinct mechanisms: TRAMP4-1 plus TRAMP4-2 bind proximal to the TSS, whereas Trf5 targets the 3' end, probably in the TRAMP5-1 complex. In consequence, Air1 hits within the mRNA are split between TSS and the pA site. Since the data are normalized across each mRNAs, this results in a reduced 3' peak for Air1 relative to Trf5. There is however a 3' peak for Air1 in cluster 1 relative to, for example, cluster 4. This is described in the revised text and shown in new Figure S7.

Minor comments:

Figure 1A: This panel is too crowded and the reader does not know where to focus on. I would suggest to simplify the color scheme of the alignment itself by reducing it to one colour (e.g. blue) and use shades to show different degrees of conservation. It is in my opinion not necessary to color the different amino acid by their physico-chemical properties.

The color scheme has been changed in the revised figure. In the new Figure 1, blue is used for conserved physico-chemical properties of residues between sequences and black for the non-conserved residues.

In Figure 3 A-K: Do the dark and light colored lines represent experimental replicates? This is not explained in the figure legend.

They do – we have altered the legend to figure 3 make this clearer.

Figure 6 is mislabelled. Panel E occurs twice and in for Nab2 in the column it should be panel G. Similarly, on page 10, last paragraph needs to be corrected: which includes the abundant snoRNA species (Fig. 6B – 6E), or considering only mRNAs (Fig. 6F – 6I). I assume this should reference to (Fig. 6B-G) and then (Fig. 6H-M).

We thank the referee for noting these errors. They have been corrected.

Then the authors mention" Close colocalization was seen for Air1 and Air2 (Fig. 6E and 6K), supporting the conclusion that they are functionally redundant on many substrates. However, Air1 is not shown in the figure at all. It only displays TRAMP components Trf4, Trf5 and Air2. Could the authors please clarify this.

In Figure 6 panels B to M, all peaks have been aligned to Air1 peaks and exhibited relative to the location of Air1 peaks (indicated by the black line at position 0 in the graphs).

For this analysis, datasets trimmed of low-complexity sequences were used. Numbers of hits for each position in the genome were calculated, excluding all non-RNA polymerase II transcripts and normalized to reads per millions mapped to RNA polymerase II transcripts. Clusters consisting of 15 continuous position with more than 50 hits were included. The position with the highest read count in each cluster was selected as the "peak". Air1 peaks were used as a reference file to which other protein peaks were aligned (Figs. 6B-6M and S6).

Page 12, last sentence: When bound to the exosome, Mtr4 is positioned at the entrance to the central lumen, such that RNAs can pass through the helicase domain directly into the exosome. Please cite the literature that demonstrates the Mtr4 – exosome interaction. For yeast Mtr4 it was first shown by the group of Elena Conti (Schuller et al.) and for human Mtr4 by Chris Lima's lab (Weick et al.).

We thank the referee for noting this omission. The references have been included.

Reviewer #2 (Remarks to the Author):

The manuscript entitled "Substrate Specificity of the TRAMP Nuclear Surveillance Complexes" by Delan-Forino et al. provides a novel and extensive analysis the protein-protein and protein-RNA interactions of Trf4/5-Air1/2-Mtr4 polyadenylation (TRAMP) complexes - key cofactors for the RNA exosome - in *S. cerevisiae*. In particular, proteomic analysis of Air1, Air2, Trf4, and Trf5 reveals three distinct TRAMP complexes in vivo - TRAMP4-1 (Trf4, Air2), TRAMP4-2 (Trf4, Air1), and TRAMP5 (Trf5, Air1) - that each likely associate with Mtr4 and make differential

interactions with surveillance factors, pre-mRNA processing factors, snoRNPs, and ribosome biogenesis factors. In addition, CRAC analysis (UV-crosslinking and sequence analysis of cDNA) of Air1, Air2, Trf4, Trf5, Mtr4, and Mtr4 Δ arch, an Mtr4 mutant lacking the arch domain implicated in substrate recruitment, reveals that the TRAMP complexes have distinct substrate preferences. Notably, Air1 and Trf5 show significant binding to pre-rRNA spacer regions (ETS, ITS), whereas Air2 and Trf4 show significant binding to ncRNAs (CUTs, SUTs, XUTs). Moreover, Air2 and Trf4 show enrichment at 5' promoter-proximal regions of pre-mRNAs, whereas Air1 and Trf5 show enrichment at 3' pA site-proximal regions of pre-mRNAs. Surprisingly, on many RNA substrates, binding specificity is conferred by Trf4 and Trf5 poly(A) polymerases, as opposed to the Air1 and Air2 zinc knuckle RNA binding proteins. Trf5 binding to mRNAs is also found to be important for recruitment of Mtr4 and control of mRNA levels. Overall, this work provides novel insights into the function of the TRAMP complexes and sheds some much-needed light on the substrate specificity of these critical exosome cofactors. However, at present, the manuscript requires a few additional experiments to support the results.

Major Points:

1. To support Figure 1E, the authors should validate a few of the Trf4/5 and Air1/2 interacting proteins identified by tandem-affinity purification and mass-spectrometry using purification and Western blotting. In particular, validation of the binding of Trf4, Air1, and Air2 to the surveillance factor, Xrn1, and one snoRNP protein, e.g. Snu13 or Gar1, would be important.

We do not have antibodies to these proteins and feel that their construction and testing lies out with the focus of the present paper. We note that previous publications strongly support interactions between TRAMP and pre-snoRNPs, while nuclear functions for Xrn1 have been reported in several publications, so these interactions are not unexpected. We have altered the text to make this clearer (p5 and p6).

2. In Figure 3, the high association of Trf5, Air1, and Mtr4, but not Trf4 and Air2, within the A2-A3 region of ITS1 pre-rRNA and loss of Mtr4 association upon deletion of TRF5 suggests that degradation of 23S pre-rRNA/ITS1 pre-rRNA should be more affected in *trf5 Δ and *air1 Δ cells compared to *trf4 Δ and *air2 Δ cells. The authors should test this possibility by examining the levels of 23S rRNA/ITS1 pre-rRNA in *trf5 Δ , *air1 Δ , *trf4 Δ , and *air2 Δ cells by Northern blotting. If a 23S pre-rRNA processing defect is observed in *trf5 Δ and *air1 Δ cells, the authors should test whether overexpression of Mtr4 rescues the defect.**********

Levels of pre-rRNA maturation intermediates were tested previously¹⁻³ and, indeed, it was observed that 23S was specifically targeted for oligo-adenylation by Trf5 prior to Rrp6 mediated degradation. Levels of 23S were less affected by loss of Trf4. Notably, the same behavior and specificities were observed for 20S and 27SA2. We do not expect to see clear differences between *air1 Δ and *air2 Δ due to their high redundancy.**

3. To support Figure 7, the authors should validate some the mRNA targets that increase in *trf5 Δ cells by RT-qPCR.*

Results from RT-qPCR analyses have been included in Figure S10A. These are consistent with the RNAseq data.

In addition, given that Mtr4 association with Trf5 mRNA targets is reduced in *trf5*Δ cells, suggesting decreased recruitment of Mtr4 upon loss of Trf5 leads to increased levels of mRNAs, the authors should test whether overexpression of Mtr4 can reduce the levels of the mRNAs in *trf5*Δ cells.

Mtr4 is reported to be present in significant excess over the TRAMP complexes. It also has major, TRAMP-independent targets, most notably in rRNA maturation. However, these are reported to depend on specific interactions with proteins carrying the AIM motif. It therefore seems unlikely that overexpression will strongly affect the mRNA alterations reported here.

Although not essential, it would be interesting to know whether the increase in mRNA levels observed in *trf5*Δ cells can be reduced by a catalytically inactive mutant of Trf5.

This would indeed be interesting to know, as would further investigation of the mechanism and biological significance of the altered mRNA abundance. However, we feel that this is the basis of a project that would be better reported in a subsequent paper.

Minor Points:

1. p.3-l28 - "Mtr4" should be "MTR4" to denote human protein.

Done

2. p.3-30 - "NEXT and PAXT" acronyms should be defined.

Done

3. p.5-l.30 - "Nrd1-interacting" should be "Nrd1-interacting."

Corrected

4. p.7-l.24 - "23S RNA" should be "23S rRNA."

We would prefer to retain this term, as the 23S RNA is not an rRNA or an on-pathway pre-rRNA

5. p.9-l.30 - Should there be a reference to Fig. 4R model?

We have included a call out for the panel (now Fig. 4Q)

6. p.9-l.24 - Insert comma. "Trf4, Trf5 Air1..." should be "Trf4, Trf5, Air1..."

Done

7. p.13-l.5 - "conceivably extent..." should be "conceivably extend..."

Done

8. p.13-l.6 - define "symmetry-related molecule."

We have changed this term for clarity

9. p.13-l29 - replace "effects of the *trf5*Δ on" with "effects of deletion of TRF5 on."

Done

10. Figure 1C - define units of y-axis.

There is no unit. We have explained this in the legend

Reviewer #3 (Remarks to the Author):

In this manuscript Delan-Fiorino and colleagues report the distribution of TRAMP and exosome components on different classes of RNA in vivo using a technique called CRAC (Crosslinking analysis of cDNAs), similar to iCLIP. The authors first define the composition of alternative TRAMP complexes by quantitative mass spectrometry, using mutants and different purification conditions. In the second part of the manuscript, the authors analyze the RNA occupancy of these factors in wild type and mutant conditions. Based on these analyses, they propose a model to explain the specificity of TRAMP binding to its different substrates. They also report the unexpected binding of Trf5 to the 3' end of a set of mRNAs.

In spite of the large amount of work that it represents, this study falls short of providing the significant conceptual and mechanistic advance that would be required for publication in Nature Comm. The work remains highly descriptive, with some conclusions that are suggested but not proven by the data. The finding that Trf5 occupancy peaks at the 3' end of some mRNAs is novel and potentially very interesting, but it is poorly exploited in terms of mechanistic analysis.

Specific points:

- For all CRAC analysis using mutants (e.g. Trf4 Δ arch or Mtr4 in strains deleted for other TRAMP components), it should be shown that the abundance of the protein of interest is not altered by the mutation or the mutant background. Ideally in the same CRAC experiment

Western blots have been included in the revised version of Figure S1. Significant changes in abundance were not observed.

We also note that it was reported⁴ that *air2* Δ does not affect levels of Trf4 or Trf5, and that *trf4* Δ does not affect levels of Trf5.

- In figure 3 the authors show the effect of deleting TRAMP components on the recruitment of Mtr4 to the ITS1. The authors should assess the statistical significance of the differences they see, considering the differences seen in some replicates (e.g. Trf5 and Mtr4). For instance, it appears from panel H that deletion of Trf4 leads to a reduction of roughly 50% of the Mtr4 signal (depending on the Mtr4 replicate). Is this significant?

The authors conclude that there is only a minor effect of deleting Trf4, but it is unclear on which basis. Same for Air1/2: deletion of either protein has the same negative effect on Mtr4 binding, still only Air1 is considered to have a significant effect: why is so ?

See response to referee 1, above. The referee is correct in noting that loss of Air2 results in defects, some of which are almost as strong as for Air1, and we mention this in the revised manuscript. A new panel has been added to Figure 3, in which total reads encompassing the region between cleavage sites A2 and A3 plus 10 flanking nts, were summed for each individual replicate. The manuscript text has been modified consequently. The average was calculated for biological replicates and standard deviations are shown to facilitate direct comparison between

different TRAMP components in wild-type and mutant backgrounds. Mtr4 binding to the A2-A3 region is reduced by:

4.4-fold for *mtr4Δarch*

3.7-fold for *air1Δ*

3.2-fold for *air2Δ*

2.0-fold for *trf4Δ*

5.8-fold for *trf5Δ*.

Our results strongly suggest that TRAMP complex stability and balance is modified when one component is deleted. We observed in Figure 1 that Trf5 binds to Air2 when Air1 is deleted, whereas almost no binding was seen in the in wild-type. We suggest that in absence of Air2, Air1 is more strongly bound by Trf4 than in the WT, reducing its availability to form TRAMP5-1 and targeting of ITS1. Deletion of Trf4 has a much milder effect than deletion of Trf5 on Mtr4 binding across A2-A3 region (2-fold for *trf4Δ* compared to 5.8-fold for *trf5Δ*). This also can be explained by the redundancy of the TRAMP components; in the absence of Trf4, some Trf4-binding sites will be bound by TRF5, resulting in a loss in targeting of Trf5-specific substrates. Were Air1 more important than Trf5 for recruitment, only a mild decrease of Mtr4 binding in *trf5Δ* (and *air2Δ*) should be observed since TRAMP4-1 should then replace it; however, this is not the case.

Air2 and Trf4 do not bind this region at all (panels B-C), so it is unclear how they could be involved in Mtr4 recruitment. Still, upon their deletion, Mtr4 recruitment is affected, in the case of Air2 to similar levels as for deletion of Air1.

As noted above, loss of Trf4 or Air2 likely leads to recruitment of Trf5 or Air1 to “ectopic” sites, making TRAMP5-1 less available for ITS1 processing.

The conclusion that Trf5-Air1 recruits Mtr4 to ITS1 (title of the section) is not justified in the light of the effects seen with Air2 and Trf4. Moreover, a decrease in RNA occupancy is not formal proof of a recruitment defect.

We have no other way to assess specific recruitment to ITS1. Crosslinking of Mtr4 to pre-rRNA, and its loss in mutants, has previously be taken as demonstrating a recruitment defect (see for example ⁵)

- FigS4: The significance of this figure is unclear. The authors compare the effect of deleting the arch domain to the effect of deleting Trf5 on the recruitment of Mtr4 (panel A). In a similar plot (C) they also look at the effect of *trf4Δ* versus *trf5Δ*. The scatter plots do not show clear correlations between the datasets. The authors conclude that $\Delta arch$ has a stronger effect than *trf5Δ* and *trf4Δ* has a stronger effect than *trf5Δ*, but there is only a qualitative, visual assessment of these differences. There might be more genes responsive to $\Delta arch$ than $\Delta trf5$ (S4A) but this is not evaluated statistically. Also by visual assessment there are as many, or possibly more genes for which the binding of Mtr4 is actually increased in a $\Delta arch$ (panel A, left of the y-axis) and certainly more genes with increased Mtr4 binding in the absence of Trf5 (panel C, below the x-axis): why these are ignored ?

The aim of this analysis was to study how loss of Trf4, Trf5 and the Mtr4 Arch domain correlate on mRNA substrates, in order to assess shared or individual dependencies. However, we agree that this analysis is difficult to follow and makes only a limited contribution to the MS. We have therefore removed it from the revised MS.

- The metagene analyses shown in figure 4 should be repeated using the median instead of the mean, to avoid effects that might be due to outliers. This, for instance, occurs for the 3'-end peak of Air1 that only is detected at a minority of genes (maybe 10%, group 2). Also, the strong difference between the replicates is a major concern here.

The metagene analysis was not generated using the mean. All reads for individual repeats are aligned either to TSS or pA site and hits are simply summed up at each position. Two individual biological repeats are shown in light and dark color.

- The 3' end occupancy peak observed for Trf5 is very convincing (although here too the replicates are very different). This potentially important finding is not really investigated further, except for one RNAseq analysis that detects a decrease in 3'-end reads for the genes of group 2: but what are these 3'-end truncated transcripts ? Different isoforms because of alternative polyadenylation ? Products of partial degradation ? These are important questions that, regrettably, are not addressed.

These are indeed important questions that deserve further investigation. They would, however, involve a considerable amount of work and we feel that they lie beyond the scope of the current paper.

- The conclusion that substrate specificity is provided by Trf4 and Trf5 is only based on occupancy effects in mutants, which can be very indirect. At least some in vitro RNA binding experiments should be provided to support these conclusions: are Trf4 and Trf5 able to bind RNA specifically and independently of Air1/2 ?.

This is an interesting question, but we are unclear what the referee intends by the request that we demonstrate "specific" binding by Trf4 and Trf5. We have previously shown that Trf4 and Trf5 are active RNA-modifying enzymes, so conditions could presumably be found in which RNA binding is observed. However, we do not anticipate that specific recognition will be based primarily on sequence recognition. We ran MIMIC analyses, but these failed to identify any clear consensus sequence. We mention this in the revised text. Our prediction is that specificity arises, at least in part, through colocalization of multiple proteins; Figures 2C, 5 and 6 are a first attempt to address this model. We have altered the text to make this clearer.

We also note that the double *air1*Δ *air2*Δ mutant is viable (though greatly impaired in growth) while the double *trf4*Δ *trf5*Δ is inviable, suggesting Trf4 and/or Trf5 is still able to bind RNA to some extent without an Air counterpart.

Minor points:

- Detection of Xrn1 association with TRAMP, even though RNase sensitive, is surprising. The authors should consider the possibility that Xrn1 associates in the extract, after cell breakage.

Like the referee, we were initially surprised by this result. However, while this paper was under review, it was reported that Xrn1 directly alters RNAPII transcription elongation (Begley, Corzo et al., 2019). This is consistent with previous reports of coprecipitation of Xrn1 with RNAPII (Harlen & Churchman, 2017) a nuclear function for Xrn1⁶⁻⁹. Interactions between Xrn1 and the transcription machinery and/or the nascent transcript presumably underlies the RNase-sensitive interactions detected here. We have altered the text to make this clearer.

Notably, we observed that Xrn1 was recovered across rRNA, especially at 3' end of 20S.

References

Begley V, Corzo D, Jordán-Pla A, Cuevas-Bermúdez A, Miguel-Jiménez Ld, Pérez-Aguado D, Machuca-Ostos M, Navarro F, Chávez MJ, Pérez-Ortín JE, Chávez S (2019) The mRNA degradation factor Xrn1 regulates transcription elongation in parallel to Ccr4. *Nucleic acids research* 47: 9524-9541

Harlen KM, Churchman LS (2017) Subgenic Pol II interactomes identify region-specific transcription elongation regulators. *Molecular systems biology* 13: 900-900

[OB]

- 1 Houseley, J. & Tollervey, D. Yeast Trf5p is a nuclear poly(A) polymerase. *EMBO Rep.* **7**, 205-211 (2006).
- 2 LaCava, J. et al. RNA degradation by the exosome is promoted by a nuclear polyadenylation complex. *Cell* **21**, 713-724 (2005).
- 3 Wery, M., Ruidant, S., Schillewaert, S., Lepore, N. & Lafontaine, D. L. The nuclear poly(A) polymerase and Exosome cofactor Trf5 is recruited cotranscriptionally to nucleolar surveillance. *RNA* **15**, 406-419 (2009).
- 4 Stuparevic, I., Mosrin-Huaman, C., Hervouet-Coste, N., Remenaric, M. & Rahmouni, A. R. Cotranscriptional Recruitment of RNA Exosome Cofactors Rrp47p and Mpp6p and Two Distinct Trf-Air-Mtr4 Polyadenylation (TRAMP) Complexes Assists the Exonuclease Rrp6p in the Targeting and Degradation of an Aberrant Messenger Ribonucleoprotein Particle (mRNP) in Yeast. *Journal of Biological Chemistry* **288**, 31816-31829 (2013).
- 5 Thoms, M. et al. The Exosome Is Recruited to RNA Substrates through Specific Adaptor Proteins. *Cell* **162**, 1029-1038, doi:10.1016/j.cell.2015.07.060 (2015).
- 6 Petfalski, E., Dandekar, T., Henry, Y. & Tollervey, D. Processing of the precursors to small nucleolar RNAs and rRNAs requires common components. *Mol. Cell. Biol.* **18**, 1181-1189 (1998).
- 7 Haimovich, G. et al. Gene Expression Is Circular: Factors for mRNA Degradation Also Foster mRNA Synthesis. *Cell* **153**, 1000-1011 (2013).
- 8 Blasco-Moreno, B. et al. The exonuclease Xrn1 activates transcription and translation of mRNAs encoding membrane proteins. *Nature communications* **10**, 1298-1298 (2019).

- 9 Stevens, A., Hsu, C. L., Isham, K. R. & Larimer, F. W. Fragments of the internal transcribed spacer 1 of pre-rRNA accumulate in *Saccharomyces cerevisiae* lacking 5'-3' exoribonuclease 1. *J. Bacteriol.* **173**, 7024-7028 (1991).

Reviewers' comments:

Reviewer #1 (Remarks to the Author):

The authors have improved the manuscript, especially the presentation of the key figures, as suggested. Now it is easier to compare the effects of different mutants.

Unfortunately, the domain swapping experiments of Air1 and Air2 did not work. But the deletion of CTD of Trf5 provides interesting insights into potential targeting specificity.

The manuscript attempts to gain insight into the difference in substrate specificity as well as the mechanism of TRAMP recruitment.

I was hoping for some additional experiments that could more mechanistic insights of some aspects of this work.

It contains a large amount of data, but in its current form it is still very descriptive and I was hoping for some deeper mechanistic insights.

For example:

The most important experiment for me is the one in figure 3, which concludes that Trf5 is mainly responsible for the recognition of the 5' -A3-site and that Air1 only contributes to a smaller extent. The Trf5/Air1 complex then recruits Mtr4. The deletion of the TRF5-CTD reduces targeting to this site and thus supports this.

Deletion of Air1, Air2, and Trf4 also affects Mtr4 recruitment. Here is the interpretation that these are indirect effects caused by the change in the relative distribution of the TRAMP variants. This is a reasonable hypothesis. In my opinion, the following experiment should clarify this conclusively. The deletion of AIR1 results in the formation of Trf5/Air2, according to figure 1C, which otherwise does not exist. Removal of AIR1 should thus lead to artificial recruitment of Air2 (Trf5-Air2) to the 5' -A3-site if Trf5 is responsible for the recruitment to the 5' -A3-site.

The authors tested Mtr4 recruitment in all the genetic backgrounds (dAIR1/2 and dTRF4/5). But as I said, the best experiment would be to test increased Air2 binding to 5' -A3-site in the AIR1 deletion strain.

Minor Comments:

Figure 7C&D: Figure aesthetics – please shift the x-axis up or down, y-axis left/right, so that the data lines do not cross the values of the axis

p7, second last line: 'Tr4' should read 'Trf4'

p13, Discussion, second paragraph: '... 14 different publications, SDG...' should read ',SGD'

p15, Section strains: 'Depletion was check by western blotting' should read "Depletion was checked by western blotting'

p15, Section Western blotting

'Proteins extracted as described' should read 'proteins were extracted as described'

p15 'Yeast.])' should read 'Yeast'

p16

'Protein Alignment Should read 'Protein sequence alignment.'

Also, please cite the reference for the software used.

p17, CRAC

100 sec – please replace 'sec' by 's' as the SI unit of seconds is s.

p26, Figure legends:

1) The panel of figure 4Q is either missing or the text of the figure legend should be removed" Figure 4Q: The following text is: Correlation of relative recovery of mRNAs by Mtr4 compared to Mtr4 Δ arch and Mtr4 in the trf4 Δ strain. The top 1,500 mRNA bound by Mtr4 were used in this graph. Ratios were calculated using RPM.

2) Figure legend 4R should be 4Q then.

3) The model in 4Q shows an interaction between Nab3 and Rrp6. However, the reference demonstrating this interaction is missing (Fasken Plos Genet 2015;)

Reviewer #2 (Remarks to the Author):

The revised manuscript entitled "Substrate Specificity of the TRAMP Nuclear Surveillance Complexes" by Delan-Forino et al. is now greatly improved and suitable for publication. The authors have adequately addressed all my comments and the conclusions are novel and well-supported. This excellent work provides a wealth of new information on the substrate specificity of TRAMP exosome cofactor complexes and will be of huge interest to the field and wider research community. This work is also highly likely to impact the thinking and models on exosome cofactors in the field.

We thank the referee for careful attention in reviewing the MS.
Detailed responses are given below.

Reviewer #1 (Remarks to the Author):

The authors tested Mtr4 recruitment in all the genetic backgrounds (dAIR1/2 and dTRF4/5). But as I said, the best experiment would be to test increased Air2 binding to 5' -A3-site in the AIR1 deletion strain.

We have performed this experiment, testing both RNA binding by Air1 in an *air2Δ* strain and binding of Air2 in an *air1Δ* strain (new Figure S4). The results, particularly the distribution of Air1 and Air2 across mRNAs, strongly support the conclusion that specificity of binding is not driven by Air1 or Air2. In the absence of either Air1 or Air2, the binding of each protein is very similar, and shows the combined distribution of the two proteins alone – with peaks both in the TSS proximal region (from Air2) and at the pA site (from Air1). For ITS1, loss of Air1 does not lead to increased recruitment of Air2. However, loss of Air2 strongly depletes binding by Air1. This is accompanied with greatly increased overall binding to mRNAs – presumably reflecting increased TRAMP4-1 formation and correspondingly less TRAMP5-1 than in wild-type. The minimal effect on Air2 binding from loss of Air1 probably indicates that a Trf5-Mtr4-Air2 complex is not efficiently recruited to Trf5-specific targets. These points are discussed in the revised MS (PP 8, 9, 10 and 14).

Minor Comments:

All points have been corrected and the requested references included.

Figure 7C&D: Figure aesthetics – please shift the x-axis up or down, y-axis left/right, so that the data lines do not cross the values of the axis

p7, second last line: 'Tr4' should read 'Trf4'

p13, Discussion, second paragraph: '... 14 different publications, SDG...' should read ',SGD'

p15, Section strains: 'Depletion was check by western blotting' should read "Depletion was checked by western blotting"

*p15, Section Western blotting
'Proteins extracted as described' should read 'proteins were extracted as described'*

p15 'Yeast.]' should read 'Yeast'

*p16
'Protein Alignment Should read 'Protein sequence alignment.'
Also, please cite the reference for the software used.*

*p17, CRAC
100 sec – please replace 'sec' by 's' as the SI unit of seconds is s.*

p26, Figure legends:

1) The panel of figure 4Q is either missing or the text of the figure legend should be removed” Figure 4Q: The following text is: Correlation of relative recovery of mRNAs by Mtr4 compared to Mtr4 Δ arch and Mtr4 in the trf4 Δ strain. The top 1,500 mRNA bound by Mtr4 were used in this graph. Ratios were calculated using RPM.

2) Figure legend 4R should be 4Q then.

3) The model in 4Q shows an interaction between Nab3 and Rrp6. However, the reference demonstrating this interaction is missing (Fasken Plos Genet 2015;)